# From Token to Rhythm: A Multi-Scale Approach for ECG-Language Pretraining

**Fuying Wang** [*1]   **Jiacheng Xu** [*1]   **Lequan Yu** [1]

## Abstract

Electrocardiograms (ECGs) play a vital role in monitoring cardiac health and diagnosing heart diseases. However, traditional deep learning approaches for ECG analysis rely heavily on large-scale manual annotations, which are both time-consuming and resource-intensive to obtain. To overcome this limitation, self-supervised learning (SSL) has emerged as a promising alternative, enabling the extraction of robust ECG representations that can be efficiently transferred to various downstream tasks. While previous studies have explored SSL for ECG pretraining and multimodal ECG-language alignment, they often fail to capture the multi-scale nature of ECG signals. As a result, these methods struggle to learn generalized representations due to their inability to model the hierarchical structure of ECG data. To address this gap, we introduce MELP a novel **M**ulti-scale **E**CG-**L**anguage **P**retraining (**MELP**) model that fully leverages hierarchical supervision from ECG-text pairs. MELP first pretrains a cardiology-specific language model to enhance its understanding of clinical text. It then applies three levels of cross-modal supervision—at the token, beat, and rhythm levels—to align ECG signals with textual reports, capturing structured information across different time scales. We evaluate MELP on three public ECG datasets across multiple tasks, including zero-shot ECG classification, linear probing, and transfer learning. Experimental results demonstrate that MELP outperforms existing SSL methods, underscoring its effectiveness and adaptability across diverse clinical applications. Our code is available at https://github.com/HKU-MedAI/MELP.

---

[*]Equal contribution  [1]School of Computing and Data Science, The University of Hong Kong, Hong Kong SAR, China. Correspondence to: Lequan Yu <lqyu@hku.hk>.

*Proceedings of the 42nd International Conference on Machine Learning*, Vancouver, Canada. PMLR 267, 2025. Copyright 2025 by the author(s).

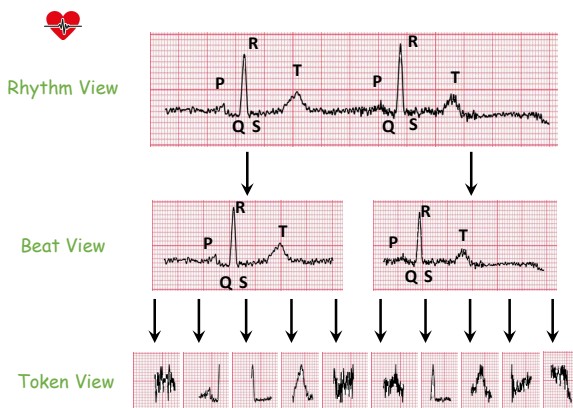

*Figure 1.* Illustration of the multi-scale view of ECG signals. **Rhythm Level**: Captures the full ECG recording, reflecting the heart's global electrical activity over time. **Beats Level**: Segments each rhythm into discrete heartbeat tokens, isolating individual cardiac cycles for localized analysis. **Token Level**: Further decomposes each heartbeat into finer-grained temporal components, enabling granular feature extraction.

## 1. Introduction

Electrocardiograms (ECGs) are widely used for monitoring cardiac health and diagnosing cardiovascular diseases. The standard 12-lead ECG records electrical activity from different perspectives, capturing both temporal and spatial characteristics of the heart's function. Advances in deep learning have significantly improved the analysis of these signals, enhancing the analysis of their underlying patterns (Yan et al., 2019; Ebrahimi et al., 2020; Siontis et al., 2021).

Self-supervised learning (SSL) has emerged as a promising solution for ECG analysis, enabling meaningful representation learning without the need for labeled data. Existing SSL approaches for ECGs rely on either contrastive (Kiyasseh et al., 2021; Oh et al., 2022; McKeen et al., 2024) or generative methods (Na et al., 2024; Hu et al., 2023; Jin et al., 2025), but most focus solely on ECG signals without leveraging complementary clinical knowledge. With the growing availability of clinical documentation, multi-modal learning—specifically ECG-language alignment—has gained attention. Recent studies (Zhao et al., 2024; Yu et al., 2024; Pham et al., 2024) have explored methods for linking ECG signals with textual interpretations. Notably, Liu

et al. (2024a) introduced MERL, a zero-shot ECG-language framework that uses contrastive learning to align ECG recordings with clinical reports. Despite these advances, existing models primarily focus on global ECG-to-text alignment, overlooking the rich, multi-scale structure of ECG signals (Figure 1). For example, the rhythm-level view captures overall temporal rhythm of the ECG, the beat-level view analyzes the characteristics of each cardiac cycle, and token-level view interprets specific waveform segments. All of these views are crucial for comprehensive ECG interpretation. To address this limitation, a more detailed approach, incorporating token-, beat-, and rhythm-level representations, is essential for capturing comprehensive ECG features critical for precise diagnostics.

To address this gap, we propose **MELP**, a novel **M**ulti-scale **ECG-L**anguage **P**retraining model that fully exploits the multi-scale structure of ECGs at the token, beat, and rhythm levels, incorporating detailed cross-modal knowledge from clinical text. A key component of MELP is a dedicated pretraining stage, where we first train a cardiology-specific language model before jointly training on paired ECG-text datasets. This step enhances the model's ability to interpret medical terminology and align textual information with ECG signals. To capture fine-grained modality interactions at the token level, we introduce an ECG captioning task, enabling the model to generate descriptive representations of short waveform segments. At the beat level, we extract heartbeat embeddings from token representations and sentence embeddings from word tokens, applying a contrastive learning objective to align beats with their corresponding clinical descriptions. Finally, at the rhythm level, we incorporate a global contrastive loss to learn robust representations of full ECG recordings. This multi-scale approach bridges the gap between raw ECG signals and their clinical interpretations, allowing MELP to learn highly transferable representations applicable across a range of tasks. We evaluate MELP on three public ECG datasets, demonstrating its superiority over existing self-supervised and multi-modal models in zero-shot classification, linear probing, and cross-institutional transfer learning.

In summary, our contributions are threefold:

- A novel Multi-scale ECG-Language Pretraining model (MELP) that hierarchically integrates clinical text knowledge for improved ECG representation learning.

- A structured pretraining framework with explicit cross-modal supervision at three clinically meaningful levels: token, beat, and rhythm.

- Comprehensive evaluation on three public ECG datasets, achieving state-of-the-art performance in zero-shot classification, linear probing, and cross-domain transfer learning.

## 2. Related Works

### 2.1. ECG Representation Learning

Self-supervised learning (SSL) has demonstrated significant efficacy in leveraging unlabeled data across diverse domains, including natural language processing (Devlin, 2018; Dong et al., 2019; He et al., 2020b), computer vision (Wu et al., 2018; Grill et al., 2020; Caron et al., 2021), and time-series analysis (Yue et al., 2022; Nie et al., 2022; Eldele et al., 2021). Recently, SSL has been extended to electrocardiogram (ECG) signal analysis, enabling robust representation learning for downstream pathology detection tasks (Baevski et al., 2020; Gopal et al., 2021). Existing methodologies in ECG SSL primarily fall into two categories: *contrastive* and *generative* approaches.

Contrastive methods (Sangha et al., 2024), currently the dominant paradigm, aim to maximize similarity between representations of augmented views of the same instance (positive pairs) while minimizing similarity with unrelated instances (negative pairs). Widely adopted frameworks such as SimCLR (Chen et al., 2020) and MoCo (He et al., 2020a) have inspired ECG-specific adaptations. For example, Kiyasseh et al. (Kiyasseh et al., 2021) employ tailored signal augmentations (e.g., lead masking, noise) to generate positive ECG pairs for contrastive training.

Generative approaches, such as ST-MEM (Na et al., 2024) and HeartLang (Jin et al., 2025), learn representations by reconstructing masked portions of ECG signals. These methods (Yu et al., 2023) typically occlude temporal segments (e.g., P-waves, T-waves) or entire beats and train models to recover the original waveform. HeartLang further introduces a tokenization strategy that segments ECG recordings by detecting QRS complexes, mapping these physiological events to discrete embeddings via a trainable codebook.

Recent efforts (Oh et al., 2022; McKeen et al., 2024; Song et al., 2024) combine contrastive and generative objectives to develop ECG foundation models. These frameworks are pre-trained on heterogeneous ECG datasets to learn generalizable representations. While promising, existing methods operate solely on uni-modal ECG data, overlooking the rich semantic correlations between ECG signals and clinical text reports, which is a limitation our work explicitly addresses.

### 2.2. ECG-Language Pretraining

Multi-modal representation learning aims to integrate information from diverse modalities by aligning their respective representations. Notable prior works, such as CLIP (Radford et al., 2021), employ contrastive methods to align image and text data. However, relatively few studies (Li et al., 2024; Han et al., 2024; Tian et al., 2024) have explored multi-modal representation learning in the context

of ECG data analysis. For instance, ESI (Yu et al., 2024) leverages a Large Language Model (LLM) to reinterpret electronic health record (EHR) data and guide ECG representation learning, while MERL (Liu et al., 2024a) proposes a framework that aligns ECG signals with clinical text reports to achieve strong zero-shot classification performance. Additional efforts, such as C-MELT (Pham et al., 2024), adopt generative approaches for both modalities—combining Masked Language Modeling (MLM) with Masked ECG Modeling (MEM)—and incorporate cross-modal contrastive learning for ECG-text alignment. Similarly, ECG-Chat (Zhao et al., 2024), inspired by the CoCa architecture (Yu et al., 2022) (a state-of-the-art contrastive framework), aligns ECG recordings with textual reports.

Despite these advances, existing models predominantly focus on deriving a global representation from the whole ECG recordings, overlooking fine-grained local patterns. Consequently, they do not effectively capture task-specific features at localized intervals, potentially limiting their adaptability to diverse downstream applications.

## 3. Method

### 3.1. Overview

Figure 2 provides an overview of MELP which learns generalized ECG representations through multi-scale cross-modal supervision in a self-supervised manner. We begin by introducing cardiology language pretraining strategies (Section 3.2) to equip the text encoders with domain-specific knowledge. Next, we present our multimodal pretraining framework in Section 3.3, which leverages fine-grained cross-modal supervision to align ECG and text modalities effectively. Finally, we describe the evaluation protocol for transferring the pretrained framework to downstream tasks in Section 3.4.

### 3.2. Cardiology Language Pretraining

As witnessed in the prior work (Boecking et al., 2022) in medical vision-language pretraining, the language plays a significant role in learning generalized visual representations. To maximize the language model's utility for cardiology, we pretrain a text encoder using a cardiology-focused corpus, building on the MED-CPT query encoder (Jin et al., 2023). The corpus is curated from three sources: cardiology-related data from PubMed[1], Wikipedia[2], and the MIMIC-IV-ECG training set (Gow et al., 2023), following the approach of HeartBERT (Gwon et al., 2024). We use the masked language modeling objective (Devlin, 2018) to pretrain the model. The details of pretraining the cardiology language model is presented in Appendix A.

---

[1] https://www.ncbi.nlm.nih.gov/home/develop/api/
[2] https://en.wikipedia.org/wiki/Wikipedia

### 3.3. Multimodal Pretraining

**Motivation.** Cardiologists interpret ECG signals in a hierarchical manner, analyzing features at multiple scales—from individual waveform components (tokens) to heartbeats (beats) and overall rhythm. This hierarchical approach forms the basis for many clinical diagnostic criteria. We observe that all three levels are essential for accurate ECG interpretation, as illustrated by the following examples (More examples are provided in Appendix B):

- Token-level: In diagnosing Atrial Fibrillation, clinicians look for absence of P waves and QRS duration usually $<120ms$. These features are localized within short waveform segments, corresponding to the token level. (LB, 2011)

- Beat-level: For sinus rhythm, each QRS complex should be preceded by a normal P wave True. This criteria requires to analyze two waveform components together across the entire heart beat. (Mattu et al., 2019)

- Rhythm-level: Diagnosing Left Anterior Fascicular Block (LAFB) involves identifying left axis deviation—e.g., negative deflections in leads II, III, and aVF, and positive deflections in leads I and aVL (Mattu et al., 2019). This assessment depends on recognizing patterns over the entire ECG lead.

Inspired by this structured approach, our model, MELP, integrates three levels of cross-modal supervision for ECG-language pretraining. Formally, we define the ECG encoder as $f(\cdot; \theta)$ and a text encoder $g(\cdot; \phi)$, where the ECG encoder is random initialized and the text encoder is initialized from the pretrained language model (Sec. 3.2). Given an ECG sample $X$ and its paired ECG report $T$, we omit sample index ($i$) for clarity.

**Token-view: Learning to Generate Captions.** To leverage the benefits of multimodal generative pretraining, we adopt an encoder-decoder architecture for ECG-language learning. The text decoder is designed to generate ECG reports with fine-grained detail, predicting tokenized text autoregressively based on ECG token embeddings. For each ECG sample $X$, our ECG encoder $f$ produces token-level ECG embedding $E \in \mathbb{R}^{L_t \times D}$, where $L_t$ is the number of ECG tokens and $D$ the feature dimension. To summarize these embeddings, we use an attention pooler (Yu et al., 2022) with 128 learnable query tokens, resulting in $\tilde{E} \in \mathbb{R}^{128 \times D}$. The attention pooler consists of a single multi-head attention layer, where the encoder output serves as both keys and values. Following a GPT-style autoregressive framework, the text decoder maximizes the conditional likelihood of the paired report through next-word prediction. For a token sequence $T = (< \text{BOS} >, w_1, ..., w_N, < \text{EOS} >)$, the text

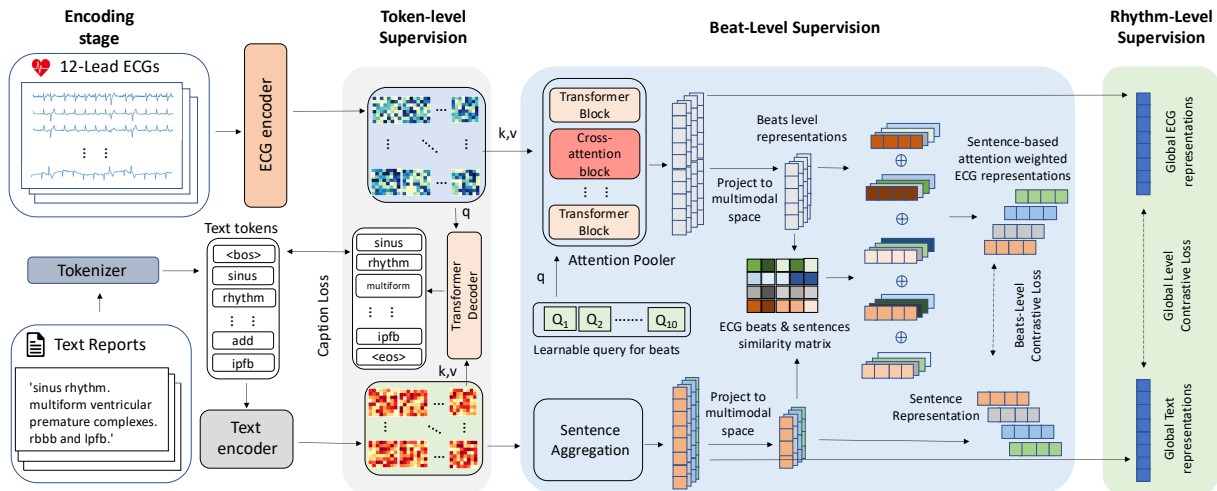

*Figure 2.* **Overview of MELP**: MELP incorporates three levels of supervision—token, beat, and rhythm—to guide ECG-language pretraining. At the token level, fine-grained ECG representations serve as queries for a transformer decoder, which reconstructs paired ECG reports using a captioning loss ($\mathcal{L}_{\text{LM}}$). At the beat level, token-level ECG features are aggregated into beat-level representations via an attention pooling layer, while text representations are grouped into sentence-level embeddings. A similarity matrix between ECG beats and text sentences is used to reweight these embeddings, optimizing a beat-level contrastive loss ($\mathcal{L}_{\text{Local}}$). At the rhythm level, beat-level ECG features and sentence-level text embeddings are further aggregated through average pooling to generate global representations, which are optimized using a global contrastive loss ($\mathcal{L}_{\text{g}}$).

decoder optimizes:

$$\mathcal{L}_{\text{LM}}(\zeta) = -\sum_{i=1}^{N} \log p(w_i | w_{0:i-1}, \tilde{E}) \qquad (1)$$

where $\zeta$ denotes the learnable parameters of the text decoder. The text decoder is randomly initialized and trained with teacher-forcing (Williams & Zipser, 1989) to enhance computational efficiency and learning speed. An additional advantage of this generative pretraining approach is its flexibility in adapting to downstream tasks, such as ECG report generation and ECG Question Answering (ECG-QA).

**Discussion.** While some ECG reports provide high-level rhythm summaries (e.g., "sinus rhythm"), many also include detailed descriptions of waveform-level abnormalities. Examples of such observations are provided in Appendix B. Besides, high-level findings, like "sinus rhythm", still depend on low-level indicators such as P wave consistency and PR interval regularity. By providing full waveform features to the decoder, token-level pretraining allow the model to learn these relationships and generate reports with varying levels of granularity. It may encourage the model to analyze local features and implicitly learn these indicators.

**Beat view: Heart Beat-Sentence Alignment.** ECG signals can be segmented into individual heart beats, allowing for more interpretable cardiological analysis (Jin et al., 2025). However, strictly relying on R-peak detection may disrupt inter-beat morphological information and temporal relationships. To mitigate this, we aggregate beat embeddings in the

latent space using an attention pooler. Specifically, we introduce 10 learnable tokens to hierarchically summarize beats within a 10-second ECG segment. While we use a default setting of 10 tokens in the main manuscript, an empirical analysis presented in Appendix C.2 shows that this is not necessarily the optimal choice. We leave the exploration of more adaptive and clinically informed heartbeat alignment strategies for future work.

Clinical observations suggest that transient abnormal beats often correspond to specific sentences in ECG reports. To capture this fine-grained alignment, we propose a beat-sentence matching mechanism. Let $B \in \mathbb{R}^{N_B \times D}$ represent beat embeddings obtained from the ECG encoder's attention-pooled outputs, and let $S \in \mathbb{R}^{S \times D}$ denote sentence embeddings, where each sentence embedding is computed by averaging its word tokens. A projection layer—either a linear transformation or a two-layer MLP—maps both embeddings into a shared latent space using functions $p_{\text{E}}$ (for ECG) and $p_{\text{T}}$ (for Text). For simplicity, we continue to denote the resulting embeddings as $B$ and $S$. For each sentence embedding $S(l) \in \mathbb{R}^D$, we compute an attention-weighted beat embedding $\hat{B}(l)$ using this equation:

$$\hat{B}(l) = \sum_{l=1}^{N_B} \alpha_l S(l) \qquad (2)$$

where the attention weight $\alpha_l$ is defined as:

$$\alpha(l) = \frac{\exp(\langle S(l), B(l) \rangle / \tau_1)}{\sum_{j=1}^{N_B} \exp(\langle S(l), B(j) \rangle / \tau_1)} \qquad (3)$$

Table 1. Details on the number of samples in each split for each downstream dataset.

| #. Samples | MIMIC-IV-ECG | PTBXL-Rhythm | PTBXL-Sub | PTBXL-Form | PTBXL-Super | CPSC2018 | CSN |
|---|---|---|---|---|---|---|---|
| Train | 745,447 | 16,832 | 17,084 | 7,197 | 17,084 | 4,950 | 16,546 |
| Validation | 15,171 | 2,100 | 2,146 | 901 | 2,146 | 551 | 1,860 |
| Test | - | 2,098 | 2,158 | 880 | 2,158 | 1,376 | 4,620 |

Here $\langle \cdot \rangle$ denotes cosine similarity, $\tau_1 = 0.25$ is a temperature hyperparameter. The similarity between an ECG-text pair is then computed by aggregating similarities between attention-weighted beat embeddings and the corresponding text embedding:

$$Z(X, T) = \log\Big( \sum_{l=1}^{S} \exp(\langle \hat{B}(l), S(l) \rangle)/\tau_2 \Big)^{\tau_2} \quad (4)$$

where $\tau_2 = 0.1$ is another temperature hyperparameter.

To optimize alignment, we define the local contrastive loss for a minibatch of size B as:

$$\mathcal{L}_{\text{Local}}^{e \rightarrow t} = \frac{1}{B} \sum_{i=1}^{B} -\log\Big( \frac{\exp(Z(X_i, T_i))/\tau_2}{\sum_{k=1}^{B} \exp(Z(X_i, T_k))/\tau_2} \Big) \quad (5)$$

where $i$ is the sample index.

The total local loss is then given by: $\mathcal{L}_{\text{Local}} = \frac{1}{2}(\mathcal{L}_{\text{Local}}^{e \rightarrow t} + \mathcal{L}_{\text{Local}}^{t \rightarrow e})$. This loss function ensures robust heartbeat-sentence alignment, preserving both local and global ECG-text relationships.

**Rhythm view: ECG-Report Alignment.** Inspired by the success of CLIP (Radford et al., 2021), contrastive loss has proven effective in learning transferable multimodal representations. Following this principle, we introduce an instance-level contrastive loss to establish high-level cross-modal supervision between ECG signals and text reports.

To obtain global representations, we compute the ECG embedding by averaging all beat embeddings, while the text embedding is represented by the [CLS] token. For the $i$-th ECG-text pair, we denote these global embeddings as $X_i^g$ (ECG) and $T_i^g$ (text). Their similarity is computed using cosine similarity as $\langle X_i^g, T_i^g \rangle$. The global alignment loss $\mathcal{L}_{\text{g}} = \frac{1}{2}(\mathcal{L}_{\text{g}}^{e \rightarrow t} + \mathcal{L}_{\text{g}}^{t \rightarrow e})$. The $\mathcal{L}_{\text{g}}^{e \rightarrow t}$ is defined following the InfoNCE formulation (Oord et al., 2018):

$$\mathcal{L}_{\text{g}}^{e \rightarrow t} = -\frac{1}{B} \log \frac{\exp(\langle X_i^g, T_i^g \rangle/\tau)}{\sum_{j=1}^{B} \exp(\langle X_i^g, T_j^g \rangle/\tau)} \quad (6)$$

where $\tau$ is a learnable temperature hyperparameter, and B denotes the batch size.

**Overall Pretraining.** Our framework is trained by jointly optimizing three loss functions:

$$\mathcal{L} = \mathcal{L}_{\text{g}} + \lambda_1 * \mathcal{L}_{\text{LM}} + \lambda_2 * \mathcal{L}_{\text{Local}} \quad (7)$$

where $\lambda_1$ and $\lambda_2$ control the contributions of the language modeling and local alignment losses, respectively. Based on empirical results, we set $\lambda_1 = 2$ and $\lambda_2 = 0.2$ for pretraining. The experimental results can be found in Sec. 4.3.5.

### 3.4. Transferring into Downstream Tasks

For zero-shot evaluation, we directly use the global embeddings $X_i^g$ (ECG) and $T_i^g$ (Text) for retrieval-based tasks.

For fine-tuning, we first extract beat-level embeddings before the projection layer $p_E$. These embeddings are then aggregated via average pooling to obtain a global feature vector. A linear classification layer is added on top to generate per-class predictions.

In the linear probing setup, we freeze the entire network and train only the linear layer.

## 4. Experiments

### 4.1. Experimental Setup

#### 4.1.1. PRETRAINING TASK

**Pretraining Dataset.** For the pretraining stage, we utilize the *MIMIC-IV-ECG v1.0 database* (Gow et al., 2023), comprising 800,035 ECG recordings from 161,352 unique patients. Each recording consists of a 10-second waveform sampled at 500 Hz. The database provides multimodal alignment through clinical text reports, with up to 18 textual reports paired with each ECG recording. We adopt preprocessing protocols adapted from (Liu et al., 2024a), including text normalization (e.g., lowercase conversion, punctuation removal, and special character elimination). To ensure data quality, we exclude: (1) ECG recordings containing empty or NaN (Not-a-Number) values, (2) text reports with fewer than four tokens, and (3) ECG samples lacking paired textual annotations. This rigorous curation process results in a final dataset of 760,618 high-quality ECG-text pairs.

**Implementation Details.** The ECG encoder is based on the Wav2Vec 2.0 architecture (Baevski et al., 2020), which integrates a multi-layer convolutional neural network (CNN) feature extractor with a transformer encoder. The ablation results justifying our selection of the Wav2Vec 2.0 architecture as the ECG encoder are included in Section 4.3.3. We use the AdamW optimizer with an initial learning rate of 2e-4, a weight decay of 0.2, and a cosine annealing learning

*Table 2.* Linear probing performance (AUC [%]) of MELP and baseline models across multiple datasets. Results are reported for different training data proportions (1%, 10%, and 100%). The best and second-best results are highlighted in bold and underlined, respectively.

| Methods | PTBXL-Rhythm | | | PTBXL-Sub | | | PTBXL-Form | | | PTBXL-Super | | | CPSC2018 | | | CSN | | |
|---|---|---|---|---|---|---|---|---|---|---|---|---|---|---|---|---|---|---|
| Training ratio | 1% | 10% | 100% | 1% | 10% | 100% | 1% | 10% | 100% | 1% | 10% | 100% | 1% | 10% | 100% | 1% | 10% | 100% |
| SimCLR (Chen et al., 2020) | 51.41 | 69.44 | 77.73 | 60.84 | 68.27 | 73.39 | 54.98 | 56.97 | 62.52 | 63.41 | 69.77 | 73.53 | 59.78 | 68.52 | 76.54 | 59.02 | 67.26 | 73.20 |
| BYOL (Grill et al., 2020) | 41.99 | 74.40 | 77.17 | 57.16 | 67.44 | 71.64 | 48.73 | 61.63 | 70.82 | 71.70 | 73.83 | 76.45 | 60.88 | 74.42 | 78.75 | 54.20 | 71.92 | 74.69 |
| BarlowTwins (Zbontar et al., 2021) | 50.12 | 73.54 | 77.62 | 62.57 | 70.84 | 74.34 | 52.12 | 60.39 | 66.14 | 72.87 | 75.96 | 78.41 | 55.12 | 72.75 | 78.39 | 60.72 | 71.64 | 77.43 |
| MoCo-v3 (Chen et al., 2021) | 51.38 | 71.66 | 74.33 | 55.88 | 69.21 | 76.69 | 50.32 | 63.71 | 71.31 | 73.19 | 76.65 | 78.26 | 62.13 | 76.74 | 75.29 | 54.61 | 74.26 | 77.68 |
| SimSiam (Chen & He, 2021) | 49.30 | 69.47 | 75.92 | 62.52 | 69.31 | 76.38 | 55.16 | 62.91 | 71.31 | 73.15 | 72.70 | 75.63 | 58.35 | 72.89 | 75.31 | 58.25 | 68.61 | 77.41 |
| TS-TCC (Eldele et al., 2021) | 43.34 | 69.48 | 78.23 | 53.54 | 66.98 | 77.87 | 48.04 | 61.79 | 71.18 | 70.73 | 75.88 | 78.91 | 57.07 | 73.62 | 78.72 | 55.26 | 68.48 | 76.79 |
| CLOCS (Kiyasseh et al., 2021) | 47.19 | 71.88 | 76.31 | 57.94 | 72.55 | 71.64 | 51.97 | 57.79 | 72.65 | 68.94 | 73.36 | 76.31 | 59.59 | 77.78 | 77.49 | 54.38 | 71.93 | 76.13 |
| Wav2Vec 2.0 + CMSC + RLM (Oh et al., 2022) | 76.24 | 86.34 | 92.05 | 69.10 | 80.71 | 85.01 | 52.72 | 67.81 | 80.72 | 81.15 | 84.88 | 85.53 | 75.70 | 88.16 | 92.61 | 65.65 | 78.82 | 87.87 |
| ASTCL (Wang et al., 2024) | 52.38 | 71.98 | 76.05 | 61.86 | 68.77 | 76.51 | 44.14 | 60.93 | 66.99 | 72.51 | 77.31 | 81.02 | 57.90 | 77.01 | 79.51 | 56.40 | 70.87 | 75.79 |
| CRT (Zhang et al., 2023) | 47.44 | 73.52 | 74.41 | 61.98 | 70.82 | 78.67 | 46.41 | 59.49 | 68.73 | 69.68 | 78.24 | 77.24 | 58.01 | 76.43 | 82.03 | 56.21 | 73.70 | 78.80 |
| ECGFM (McKeen et al., 2024) | 81.45 | 91.59 | 92.70 | 73.24 | 81.91 | 86.07 | 60.95 | 74.99 | 85.54 | 78.67 | 84.80 | 86.47 | 82.18 | 89.52 | 93.26 | 71.51 | 83.17 | 88.89 |
| ST-MEM (Na et al., 2024) | 51.12 | 65.44 | 74.85 | 54.12 | 57.86 | 63.59 | 55.71 | 59.99 | 66.07 | 61.12 | 66.87 | 71.36 | 56.69 | 63.32 | 70.39 | 59.77 | 66.87 | 71.36 |
| HeartLang (Jin et al., 2025) | 62.08 | 76.22 | 90.34 | 64.68 | 79.34 | **88.91** | 58.70 | 63.99 | 80.23 | 78.94 | 85.59 | 87.52 | 60.44 | 66.26 | 77.87 | 57.94 | 68.93 | 82.49 |
| MERL (Liu et al., 2024a) | 53.33 | 82.88 | 88.34 | 64.90 | 80.56 | 84.72 | 58.26 | 72.43 | 79.65 | 82.39 | 86.27 | **88.67** | 70.33 | 85.32 | 90.57 | 66.60 | 82.74 | 87.95 |
| MELP (Ours) | **88.83** | **94.65** | **96.91** | **79.22** | **84.40** | 87.46 | **63.41** | **76.71** | 83.30 | **85.82** | **87.61** | 87.87 | **88.54** | **91.75** | **94.32** | **78.25** | **84.83** | **90.17** |

*Table 3.* Zero-shot classification performance (AUC [%]) of MELP and baseline models across multiple datasets.

| Methods | CSN | PTBXL-Rhythm | PTBXL-Form | PTBXL-Sub | PTBXL-Super | CPSC2018 | Average |
|---|---|---|---|---|---|---|---|
| MERL (Liu et al., 2024a) | 74.4 | 78.5 | 65.9 | 75.7 | 74.2 | 82.8 | 75.3 |
| MELP (Ours) | **77.6** | **85.4** | **69.1** | **81.2** | **76.2** | **84.2** | **79.0** |
| Gains | **+3.2** | **+6.9** | **+3.2** | **+5.5** | **+2.0** | **+1.4** | **+3.7** |

rate scheduler. MELP is pretrained for 100 epochs with a per-device batch size of 64. Training is stopped early if the zero-shot prediction performance on the validation sets does not improve for five consecutive epochs. Please refer to our code for more details. All experiments are conducted on four NVIDIA GTX 3090 GPUs.

### 4.1.2. DOWNSTREAM TASKS

**Downstream Datasets.** We evaluate our pre-trained MELP across three publicly available benchmarks: PTB-XL (Wagner et al., 2020), CSN (Zheng et al., 2022), CPSC2018 (Liu et al., 2018). A summary of dataset statistics is presented in Table 1, with additional details in Appendix D. Key characteristics of these datasets are summarized below:

*PTB-XL* comprises 21,837 12-lead ECG recordings from 18,885 patients, each sampled at 500 Hz with a 10-second duration and annotated with cardiac diagnostic labels. Following the methodology of MERL (Liu et al., 2024a), we stratify the dataset into four subgroups (super, sub, form, and rhythm) for granular evaluation. Training, validation, and test splits adhere to the protocol established by (Wagner et al., 2020).

*CPSC2018* This resource contains 6,877 standard 12-lead ECG recordings sampled at 500 Hz, annotated with 9 categorical labels. We replicate the experimental configuration of MERL (Liu et al., 2024a) for consistency.

*CSN* contains 23,026 samples recorded at 500 Hz over 10-second intervals, this dataset includes 38 distinct diagnostic labels. For downstream evaluation, we adopt the train-validation-test partitioning scheme proposed by MERL (Liu et al., 2024a).

**Implementation Details.** For zero-shot evaluation, we use prompts driven by knowledge of GPT-4, following the approach in (Liu et al., 2024a). Linear probing tasks adhere strictly to the predefined train-validation-test splits from (Liu et al., 2024a). We conducted linear probing using 1%, 10% and 100% of the training data for each task following (Liu et al., 2024a). All downstream tasks are evaluated using AUROC (Area Under the Curve). We use a batch size of 128 and train for 50 epochs, with early stopping similarly triggered based on the validation AUC. Further implementation details are provided in Appendix E.

### 4.2. Quantitative Results

#### 4.2.1. EVALUATION ON LINEAR PROBING FOR ECG CLASSIFICATION

Table 2 presents the linear probing results comparing MELP with baseline methods. We evaluate our MELP against both unimodal self-supervised approaches, including TS-TCC (Eldele et al., 2021), CLOCS (Kiyasseh et al., 2021), ASTCL (Wang et al., 2024), CRT (Zhang et al., 2023), ST-MEM (Na et al., 2024), and HeartLang (Jin et al., 2025), as well as multimodal self-supervised method such as MERL (Liu et al., 2024a). MELP consistently improves classification performance across six tasks, achieving the highest accuracy in 16 out of 18 evaluation settings and ranking second in the remaining two. Its advantage is particularly evident when using only 1% of training data, where it surpasses the second-best method in AUROC by margins of +7.38%, +5.98%, +2.46%, +3.43%, +6.36% and +6.74% respectively. These results underscore MELP's effectiveness in low-data scenarios, highlighting its potential for real-world applications where labeled medical data

*Table 4.* Performance under data distribution shift. "Source Domain" refers to the dataset used for linear probing with the frozen pre-trained ECG encoder, while "Target Domain" represents the corresponding test set. The **Best** and Second-best results are shown in **Bold** and underlined.

| Source Domain / Target Domain | Zero-shot | Training Ratio | PTBXL-Super | | CPSC2018 | | CSN | | Average |
|---|---|---|---|---|---|---|---|---|---|
| | | | CPSC2018 | CSN | PTBXL-Super | CSN | PTBXL-Super | CPSC2018 | |
| SimCLR (Chen et al., 2020) | ✗ | 100% | 69.62 | 73.05 | 56.65 | 66.36 | 59.74 | 62.11 | 65.22 |
| BYOL (Grill et al., 2020) | ✗ | 100% | 70.27 | 74.01 | 57.32 | 67.56 | 60.39 | 63.24 | 65.63 |
| BarlowTwins (Zbontar et al., 2021) | ✗ | 100% | 68.98 | 72.85 | 55.97 | 65.89 | 58.76 | 61.35 | 64.13 |
| MoCo-v3 (Chen et al., 2021) | ✗ | 100% | 69.41 | 73.29 | 56.54 | 66.12 | 59.82 | 62.07 | 64.21 |
| SimSiam (Chen & He, 2021) | ✗ | 100% | 70.06 | 73.92 | 57.21 | 67.48 | 60.23 | 63.09 | 65.33 |
| TS-TCC (Eldele et al., 2021) | ✗ | 100% | 71.32 | 75.16 | 58.47 | 68.34 | 61.55 | 64.48 | 66.55 |
| CLOCS (Kiyasseh et al., 2021) | ✗ | 100% | 68.79 | 72.64 | 55.86 | 65.73 | 58.69 | 61.27 | 63.83 |
| ASTCL (Wang et al., 2024) | ✗ | 100% | 69.23 | 73.18 | 56.61 | 66.27 | 59.74 | 62.12 | 64.19 |
| CRT (Zhang et al., 2023) | ✗ | 100% | 70.15 | 74.08 | 57.39 | 67.62 | 60.48 | 63.33 | 65.51 |
| ST-MEM (Na et al., 2024) | ✗ | 100% | 76.12 | **84.5** | 62.27 | 75.19 | 73.05 | 64.66 | 72.63 |
| MERL (Liu et al., 2024a) | ✓ | 0% | **88.21** | 78.01 | 76.77 | 76.56 | 74.15 | **82.86** | 79.42 |
| MELP (Ours) | ✓ | 0% | 87.75 | 74.11 | **77.89** | **80.32** | **74.67** | 82.72 | **79.58** |

*Table 5.* Ablation results of loss functions on 6 linear probing tasks. The first row indicates training with only the instance-level contrastive loss $\mathcal{L}_g$. The **Best** and Second-best results are shown in **Bold** and underlined.

| $\mathcal{L}_g$ | $\mathcal{L}_{LM}$ | $\mathcal{L}_{Local}$ | PTBXL-Rhythm | | | PTBXL-Form | | | PTBXL-Sub | | | PTBXL-Super | | | CPSC2018 | | | CSN | | | Average |
|---|---|---|---|---|---|---|---|---|---|---|---|---|---|---|---|---|---|---|---|---|---|
| | | | 1% | 10% | 100% | 1% | 10% | 100% | 1% | 10% | 100% | 1% | 10% | 100% | 1% | 10% | 100% | 1% | 10% | 100% | |
| ✓ | | | 83.78 | 88.44 | 94.98 | 57.93 | 72.14 | 82.07 | 77.32 | 81.97 | 84.36 | 84.55 | 87.24 | 87.52 | 78.52 | 87.07 | 92.57 | 75.94 | 82.04 | 86.66 | 82.51 |
| | ✓ | | 77.64 | 79.44 | 85.21 | 52.95 | 63.80 | 76.91 | 71.41 | 76.67 | 82.97 | 78.73 | 82.80 | 85.18 | 64.19 | 73.05 | 85.26 | 69.81 | 79.37 | 84.41 | 76.10 |
| | | ✓ | 81.04 | 89.88 | 96.67 | 49.81 | 67.82 | 81.41 | 66.14 | 81.38 | 84.76 | 79.94 | 87.49 | 87.73 | 64.18 | 84.08 | 93.17 | 55.89 | 79.77 | 88.79 | 78.89 |
| ✓ | ✓ | | 83.25 | 89.87 | 94.86 | 56.58 | 72.71 | 81.99 | 78.61 | 82.14 | 85.84 | 84.62 | 87.18 | 87.56 | 83.74 | 88.40 | 92.77 | 74.86 | 80.48 | 87.11 | 82.92 |
| ✓ | | ✓ | 84.36 | 88.44 | 95.29 | 57.22 | 72.07 | 82.96 | **81.20** | 82.89 | 85.42 | 84.80 | 87.25 | 87.57 | 76.97 | 86.31 | 92.26 | 73.77 | 81.43 | 81.50 | 82.32 |
| ✓ | ✓ | ✓ | **88.83** | **94.65** | **96.91** | **63.41** | **76.71** | **83.30** | 79.22 | **84.40** | **87.46** | **85.82** | **87.61** | **87.87** | **88.54** | **91.75** | **94.32** | **78.25** | **84.83** | **90.17** | **85.78** |

is scarce. With full training data (100%), MELP continues to rank first or second for all datasets , improving AUROC by +4.21%, +1.06%, and +1.28% on PTBXL-Rhythm, CPSC2018, and CSN, respectively. This demonstrates MELP's ability to achieve superior performance even with abundant labeled data, setting a strong upper bound for ECG classification.

### 4.2.2. EVALUATION ON ZERO-SHOT ECG CLASSIFICATION

We evaluate MELP's zero-shot classification performance against the multimodal baseline MERL (Liu et al., 2024a). As shown in Table 3, MELP achieves state-of-the-art results across all tasks, demonstrating significant performance improvements. Specifically, MELP demonstrates improvements of +3.2%, +6.9%, +3.2%, +5.5%, +2.0%, +1.4% on CSN, PTBXL-Rhythm, PTBXL-Form, PTBXL-Sub, PTBXL-Super, and CPSC2018 respectively. Moreover, MELP achieves an average performance gain of 3.7% over the MERL. These findings highlight MELP's effectiveness in zero-shot ECG classification, reinforcing its potential for clinical applications where evaluation on diverse downstream tasks is required without additional training.

### 4.2.3. EVALUATION ON TRANSFER LEARNING FOR ECG CLASSIFICATION

To assess robustness to domain shifts, we evaluate transfer learning performance using test datasets that differ in distribution from the pretraining data but share the same label space. For non-zero-shot self-supervised learning (SSL) baselines, we apply linear probing using 100% of the source domain training data. For zero-shot models (MERL), we follow the target-source category matching protocol outlined in Appendix E. This evaluation measures MELP's ability to generalize across diverse clinical settings. As shown in Table 4, MELP achieves the highest performance in three out of six settings and ranks second in two. Notably, it also achieves the best average performance across all six tasks. Specifically, our MELP outperforms MERL by +1.12%, +3.76%, +0.52% on the CPSC2018 → PTXBL-Super, CPSC2018 → CSN and CSN → PTXBL-Super settings, respectively. These results highlights MELP's robustness and its ability to handle distribution shifts effectively.

### 4.3. Analysis of Our Framework

#### 4.3.1. ABLATION ON MULTI-SCALE SUPERVISION

To evaluate the impact of multi-scale supervision, including token-level supervision ($\mathcal{L}_{LM}$), beat-level supervision ($\mathcal{L}_{Local}$), and rhythm-level supervision ($\mathcal{L}_g$), we conduct ablation studies over different variants. This allows us to assess each supervision level's contribution. As shown in Table 5, The first three rows present models trained with single isolated supervision (one loss each). Rows four and five correspond to variants without token-level supervision ($\mathcal{L}_{LM}$) and without beat-level supervision ($\mathcal{L}_{Local}$), respectively. The final row shows the performance of the full model. Our full model achieves the highest perfor-

*Table 6.* Ablation results of loss functions on 6 zero-shot classification tasks.

| $\mathcal{L}_{LM}$ | $\mathcal{L}_{Local}$ | PTBXL-Rhythm | PTBXL-Form | PTBXL-Sub | PTBXL-Super | CPSC2018 | CSN | Average |
|---|---|---|---|---|---|---|---|---|
| | | 79.6 | 68.7 | 77.1 | **77.2** | 83.9 | 75.0 | 76.9 |
| ✓ | | 84.1 | 67.4 | 74.0 | 74.1 | 81.9 | 76.8 | 76.4 |
| | ✓ | 82.8 | 64.4 | 79.0 | 75.9 | 81.1 | 76.9 | 76.7 |
| ✓ | ✓ | **85.4** | **69.1** | **81.2** | 76.2 | **84.2** | **77.6** | **79.0** |

*Table 7.* Ablation results of ECG encoders. We have used RLM as the augmentation technique for ECG by default. CMSC can't easily integrate into our model since it needs to split the ECG into two parts and performs contrastive learning.

| ECG encoder | PTBXL-Rhythm | | | PTBXL-Form | | | PTBXL-Sub | | | PTBXL-Super | | | CPSC2018 | | | CSN | | | Average |
|---|---|---|---|---|---|---|---|---|---|---|---|---|---|---|---|---|---|---|---|
| | 1% | 10% | 100% | 1% | 10% | 100% | 1% | 10% | 100% | 1% | 10% | 100% | 1% | 10% | 100% | 1% | 10% | 100% | |
| ResNet-18 | 85.10 | 90.11 | 94.31 | 62.82 | 73.59 | 79.23 | 75.59 | 81.72 | 85.70 | **85.84** | 86.99 | 87.24 | 83.31 | 89.75 | 93.35 | 68.79 | 82.12 | 89.71 | 83.07 |
| Wav2Vec 2.0 | **88.83** | **94.65** | **96.91** | **63.41** | **76.71** | **83.30** | **79.22** | **84.40** | **87.46** | 85.82 | **87.61** | **87.87** | **88.54** | **91.75** | **94.32** | **78.25** | **84.83** | **90.17** | **85.78** |
| Wav2Vec 2.0 + CMSC | 83.15 | 88.25 | 94.82 | 62.07 | 75.55 | 82.57 | 77.21 | 82.29 | 84.85 | 85.14 | 87.52 | 87.64 | 80.69 | 88.40 | 92.91 | 71.89 | 81.00 | 87.42 | 82.97 |

*Table 8.* Ablation results of pretrained domain-specific language model on 6 linear probing tasks.

| Text | PTBXL-Rhythm | | | PTBXL-Form | | | PTBXL-Sub | | | PTBXL-Super | | | CPSC2018 | | | CSN | | | Average |
|---|---|---|---|---|---|---|---|---|---|---|---|---|---|---|---|---|---|---|---|
| | 1% | 10% | 100% | 1% | 10% | 100% | 1% | 10% | 100% | 1% | 10% | 100% | 1% | 10% | 100% | 1% | 10% | 100% | |
| | 88.61 | **94.95** | 96.72 | **65.86** | 75.34 | 82.94 | **80.36** | 84.17 | 86.43 | **86.08** | 87.46 | 87.77 | 86.93 | 91.33 | **93.48** | 75.82 | 83.95 | 89.51 | 85.43 |
| ✓ | **88.83** | 94.65 | **96.91** | 63.41 | **76.71** | **83.30** | 79.22 | **84.40** | **87.46** | 85.82 | **87.61** | **87.87** | **88.54** | **91.75** | 94.32 | **78.25** | **84.83** | **90.17** | **85.78** |

*Table 9.* ECG report generation metrics ([%]) on 500 curated samples from PTB-XL report dataset.

| Models | Size | BLEU-1 | BLEU-4 | METEOR | ROUGE-L | BERTScore F1 |
|---|---|---|---|---|---|---|
| PULSE (Liu et al., 2024b) | 7B | 5.12 | 0.83 | **13.76** | 8.15 | 10.96 |
| MELP | 284M | **13.02** | **1.87** | 11.28 | **18.50** | **44.08** |

mance on most benchmarks, outperforming the best partial-supervision variant by an average AUROC gain of $2.86\%$. Furthermore, we observe that variants without rhythm-level supervision ($\mathcal{L}$g) perform significantly worse, while all configurations including $\mathcal{L}$g achieve AUROCs above 82%. This demonstrates that global-level multi-modal alignment is essential for effective ECG representation learning. This empirically demonstrates the importance of hierarchical supervision—integrating rhythm-level ($\mathcal{L}_{g}$), beat-level ($\mathcal{L}_{Local}$), and token-level ($\mathcal{L}_{LM}$) signals—for learning clinically meaningful ECG representations. Furthermore, we extend this analysis to zero-shot classification following the framework used in previous ablation studies. As shown in Table 6, the full model consistently outperforms ablated versions, particularly in low-data settings, with an average AUROC improvement of 2.1%. These findings highlight the critical role of multi-scale supervision in learning robust and transferable ECG representations.

### 4.3.2. ANALYSIS ON ECG REPORT GENERATION

Although MELP is primarily designed to encode generalized global ECG representations, we further evaluate its fine-grained prediction capabilities to validate the effectiveness of our multi-scale pretraining approach. Specifically, we conduct a preliminary evaluation on the ECG report generation task using the ECGBench dataset (Liu et al., 2024b), and compare our model with PULSE (Liu et al., 2024b), a multimodal baseline trained on ECG images via instruction tuning and evaluated in a zero-shot setting. We employ both standard Natural Language Generation (NLG) metrics

and BERTScore to assess the lexical and semantic quality of the generated reports. As shown in Table 9, MELP significantly outperforms PULSE, demonstrating strong capabilities in fine-grained ECG understanding and report generation. These results highlight the potential of MELP to support clinically relevant diagnostic tasks that require detailed, fine-grained predictions. Additionally, we evaluate the model on patient identification, with results reported in Table 13. A more comprehensive analysis of fine-grained prediction performance remains an important direction for future research.

### 4.3.3. ABLATION ON ECG ENCODER

Table 7 presents an ablation study evaluating the impact of different ECG encoder backbones on the pretraining performance of MELP. Specifically, we compare the Wav2Vec 2.0 (Baevski et al., 2020) architecture with the ResNet-18 backbone used in MERL (Liu et al., 2024a). In addition, we assess the effectiveness of using CMSC (Kiyasseh et al., 2021) for pretraining the ECG encoder prior to multimodal training. CMSC is a patient-specific contrastive learning method for unlabeled ECG data and is orthogonal to the choice of ECG encoder architecture. The results show that employing Wav2Vec 2.0 as the ECG encoder consistently outperforms the CMSC-based variant across all evaluation settings, with an average gain of $+3.53\%$. Furthermore, it outperforms the ResNet-18-based variant in 17 out of 18 settings, with an average improvement of $+3.63\%$.

### 4.3.4. ABLATION ON UNI-MODAL TEXT PRETRAINING

To evaluate the impact of uni-modal text pretraining, we compare the full model with a variant that omits this step. As shown in Table 8, the baseline model (first row) operates without text pretraining, while the full model (second row) incorporates it. The full model outperforms the baseline in

*Table 10.* Ablation results of hyperparameters on 6 zero-shot classification tasks.

| $\lambda_1$ | $\lambda_2$ | PTBXL-Rhythm | PTBXL-Form | PTBXL-Sub | PTBXL-Super | CPSC2018 | CSN | Average |
|---|---|---|---|---|---|---|---|---|
| 1 | 1 | 85.3 | 66.3 | 76.9 | 75.3 | 82.9 | 75.6 | 77.1 |
| 1 | 0.2 | 84.6 | 69.0 | 75.2 | 76.1 | 84.0 | 77.0 | 77.6 |
| 2 | 1 | 79.6 | 68.7 | 77.1 | 77.2 | 83.9 | 75.2 | 76.9 |
| 2 | 0.2 | **85.4** | **69.1** | **81.2** | **76.2** | **84.2** | **77.6** | **78.9** |

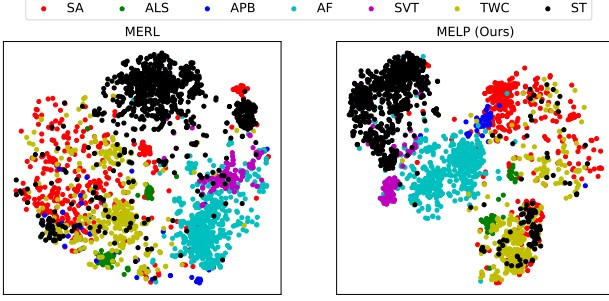

*Figure 3.* Comparison of T-SNE visualizations of the embedding space for MERL and MELP on the CSN test set.

13 out of 18 settings, achieving an average improvement of 0.35%. These results suggest that text pretraining enhances downstream generalization by improving feature separation in the joint embedding space. They also highlight the potential benefits of developing more specialized cardiology-specific language models to further improve performance.

### 4.3.5. ABLATION ON HYPERPARAMETERS

We evaluate the impact of hyperparameters $\lambda_1$ and $\lambda_2$ on zero-shot classification performance, as shown in Table 10. Overall, the performance remains consistent across four different hyperparameter configurations, demonstrating the robustness of our framework. Through a preliminary search, we identify $\lambda_1 = 2$ and $\lambda_2 = 0.2$ as the optimal setting, which we adopt as the default.

### 4.4. Qualitative Results

To analyze the learned representations, we visualize the embedding space of the CSN test set in Figure 3. Following (Liu et al., 2024a), we focus on seven common ECG abnormalities and select samples that exclusively exhibit each condition. We extract embeddings using both MERL and MELP and project them into a lower-dimensional space for visualization. The results show that MELP produces a more distinct and well-separated embedding space compared to MERL. This improved separability helps explain MELP's superior zero-shot classification performance.

## 5. Conclusion

We introduce MELP, a multimodal ECG foundation model that leverages multi-scale supervision from clinical text re-

ports to learn fine-grained representations for diverse downstream tasks. Extensive experiments on three benchmark datasets demonstrate its superior performance over existing baselines, highlighting its effectiveness in aligning ECG signals with clinical text at multiple levels of abstraction.

**Limitations and Future Directions.** Despite its strengths, MELP has two limitations. First, its token-level supervision lacks explicit clinical interpretability, limiting its applicability in explainable diagnostic settings, as discussed in Appendix B. Future work will explore leveraging external medical knowledge bases to generate clinically meaningful descriptions, thereby enabling more interpretable and informative token-level supervision. Second, the current model adopts the MedCPT architecture as its text encoder, which does not fully capitalize on the recent advances in large language models (LLMs). To address this, future research will investigate ECG instruction tuning frameworks that harness the strong generalization and reasoning capabilities of LLMs for enhanced multimodal understanding.

## Acknowledgment

This work was supported in part by the Research Grants Council of Hong Kong (27206123, C5055-24G, and T45-401/22-N), the Hong Kong Innovation and Technology Fund (ITS/273/22, ITS/274/22, and GHP/318/22GD), the National Natural Science Foundation of China (No. 62201483), and Guangdong Natural Science Fund (No. 2024A1515011875).

## Impact Statement

The pretraining of MELP is based on large-scale ECG-text pairs. In this study, we utilize the publicly available MIMIC-IV-ECG dataset, which has undergone rigorous de-identification procedures. Nonetheless, privacy considerations remain critical, especially when deploying the framework in other scenarios. Moreover, although MELP does not explicitly incorporate sensitive attributes such as gender or race during pretraining, its predictions may still reflect subtle biases related to these factors. We encourage future work to further investigate privacy implications and develop effective bias-mitigation strategies to ensure the ethical and equitable deployment of AI systems in healthcare. In addition, adapting our framework for clinical scenarios will require careful consideration of regulatory requirements.

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

# A. Details of Training Cardiology Language Model

## A.1. Cardiology Corpus Details

To extract cardiology-related data from PubMed and Wikipedia, we followed the procedure used in HeartBERT (Gwon et al., 2024). For PudMed dataset, we compiled a list of cardiology-related journal names from the SJR (Scimago Journal & Country Rank) database, along with glossaries from Aiken, NIH, and the Texas Heart Institute. These terms were used as queries to retrieve relevant content via the PudMed database API. To ensure content relevance, we only employ abstract section for PudMed dataset. For Wikipedia dataset, since it already provides information about categories and subcategories for classification, we use a top-level category called "Cardiology" as the primary category. Starting with the "Cardiology" category, we navigated through the subcategories provided by Wikipedia to collect related articles. To ensure content relevance, we only employ abstract section for PudMed dataset. This process finally resulted in a curated dataset of approximately 5.6 GB, containing 912.5 million corpus. In addition to the curated corpus introduced in Section A.1, we further incorporate ECG-related reports from the MIMIC-IV-ECG training set to enhance the model's understanding of ECG-specific clinical language.

## A.2. Training Details

Figure 4 illustrates the overall training workflow of our cardiology-specific language model pretraining. We initialize our model using the query encoder from MedCPT (Jin et al., 2023)[3], which was originally trained on PubMed search logs. To better adapt it to the cardiology domain, we further fine-tune this encoder using a masked language modeling (MLM) objective. Specifically, we apply random masking to tokens in ECG diagnostic reports and train the model to reconstruct the masked tokens based on their surrounding context. This encourages the encoder to capture nuanced, domain-specific semantics from cardiology narratives.

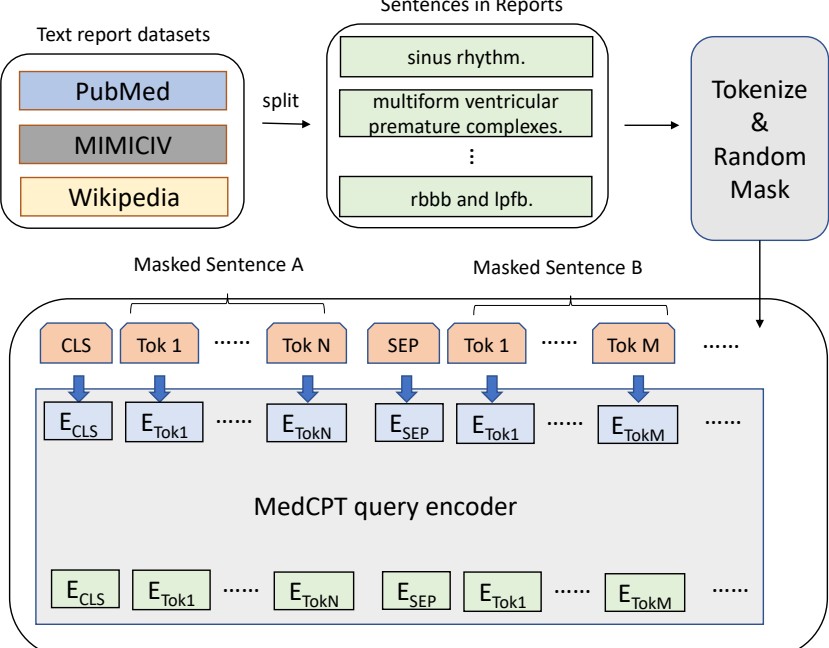

*Figure 4.* Framework for text uni-modal pretraining. Three widely-used medical datasets were combined and segmented into sentences as model inputs. Following BERT-style masking methodology (Devlin, 2018), we randomly mask portions of the input text and insert special tokens between sentences. The masked sequences are then processed through the MedCPT query encoder to generate text representations.

---

[3]https://huggingface.co/ncbi/MedCPT-Query-Encoder

# B. Discussion on Multi-Scale ECG Interpretation

While some portion of ECG reports contain high-level summaries like "sinus rhythm," many real-world diagnostic reports also include detailed references to waveform-level features. To further support this, we provide representative examples from our pretraining dataset, MIMIC-IV-ECG database (Gow et al., 2023) in Table 11. These demonstrate that detailed morphological patterns at the waveform (token) level are frequently described in the reports.

*Table 11.* Examples of ECG reports from MIMIC-IV-ECG dataset. Each row is a complete text report and those parts in bold are descriptions about fine-grained ECG details.

| |
|---|
| Regular rhythm. Lead(s) unsuitable for analysis: **V1. Q waves in inferior leads**. **T wave inversion also present**. Possible inferior infarction – age undetermined. **Anterolateral ST-T changes**. Summary: abnormal ECG. |
| Sinus tachycardia. **Short PR interval**. Borderline ECG. |
| Sinus rhythm. **Poor R wave progression** – probable normal variant. **Anterolateral T wave changes** may be due to myocardial ischemia. Abnormal ECG. |
| Atrial fibrillation. **Extensive ST-T changes are nonspecific**. Abnormal ECG. |
| Probable accelerated junctional rhythm. **Low QRS voltages in limb leads**. Abnormal ECG. |

Furthermore, even broader assessments like "sinus rhythm" are based on a set of well-established low-level criteria, such as the presence of a P wave before every QRS complex, upright P waves in leads I, II, and aVF, regular R-R intervals, consistent PR intervals, and a normal heart rate. More examples can be found in Table 12. Although these features are not explicitly mentioned in the cardiology reports in the MIMIC-IV-ECG dataset, these diagnosis are actually derived from these fine-grained waveform analysis and can be effectively captured by token-level representations during the learning process.

*Table 12.* Cardiology examples

| clinical diagnosis | ecg criteria | whether contain local descriptions |
|---|---|---|
| Left Anterior Fascicular Block (LAFB) | rS complexes in leads II, III, aVF, with small R waves and deep S waves | True (Token-level) |
| | qR complexes in leads I, aVL, with small Q waves and tall R waves | True (Token-level) |
| | Left Axis Deviation (LAD): Leads II, III and aVF are NEGATIVE; Leads I and aVL are POSITIVE | False (rhythm-level) |
| Atrial Fibrillation | Irregularly irregular rhythm | False (rhythm-level) |
| | No P waves | True (Token-level) |
| | QRS complexes usually <120ms | True (Token-level) |
| | Variable ventricular rate | False (rhythm-level) |
| Left Bundle Branch Block (LBBB) | QRS duration ≥ 120ms | True (Token-level) |
| | Dominant S wave in V1 | True (Token-level) |
| | Broad monophasic R wave in lateral leads (I, aVL, V5-6) | True (Token-level) |
| | Absence of Q waves in lateral leads | True (Token-level) |
| | Prolonged R wave peak time >60ms in leads V5-6 | True (Token-level) |
| sinus rhythm | Regular rhythm at a rate of 60-100 bpm | False (rhythm-level) |
| | Each QRS complex is preceded by a normal P wave | True (beat-level) |
| | Normal P wave axis: P waves upright in leads I and II, inverted in aVR | True (token-level) |
| | The PR interval remains constant | True (token-level) |
| | QRS complexes <100 ms wide | True (token-level) |

Building on these observations, we argue that using token-level ECG embeddings for report generation is not only appropriate but essential for capturing the full spectrum of clinically meaningful information. To clarify this motivation, we have revised the manuscript and included supporting examples that illustrate the rationale behind our design choice.

Moreover, we believe that incorporating more fine-grained textual descriptions—such as explicit references to waveform components—alongside general diagnostic terms could further strengthen our approach. These detailed references would provide richer supervision signals and more closely align with established clinical diagnostic criteria. For instance, broad terms like "sinus rhythm" could be supplemented with explicit criteria such as the presence of P waves before each QRS complex and consistent PR intervals. Integrating such detail into the training data may improve alignment between ECG signals and textual descriptions. While this direction holds promise, it is beyond the scope of the current work and is left for future exploration.

# C. Additional Experimental results

## C.1. Results for Patient Identification task

The patient identification task trains a model to recognize individual patients based on their ECG signals. It does this by learning to generate unique numerical representations for each person's ECG. The core objective is to ensure ECGs from the

*Table 13.* Zero-shot Patient identification results using Top-k recall [%]. Here we use 608 pairs for patient identification following Oh et al. (2022).

| Method | PTBXL | | |
|---|---|---|---|
| | R@1 | R@5 | R@10 |
| Wav2Vec 2.0 + CMSC + RLM (Oh et al., 2022) | 39.8 | 52.14 | 59.21 |
| ECGFM (McKeen et al., 2024) | 49.18 | 60.70 | 67.76 |
| MERL (Liu et al., 2024a) | 16.12 | 26.32 | 31.74 |
| MELP | **49.67** | **66.12** | **70.89** |

same patient produce highly similar representations, while ECGs from different patients produce distinct ones. For detailed experimental procedures, we follow the settings of (Oh et al., 2022). As shown in Table 13, our model MELPoutperforms all three baselines in top-1, top-5, and top-10 accuracy.

*Table 14.* Distribution of number of beats in the training set of MIMIC-IV-ECG.

| Beat Count | 8 | 9 | 10 | 11 | 12 | 13 | 14 | 15 | 16 | 17 | 18 | 19 | 20 | Others |
|---|---|---|---|---|---|---|---|---|---|---|---|---|---|---|
| Frequency | 18357 | 47635 | 93075 | 112010 | 112424 | 93830 | 74027 | 62150 | 47997 | 24509 | 15987 | 11606 | 8347 | 23493 |
| Percentage | 2.5% | 6.4% | 12.5% | 15.0% | 15.1% | 12.6% | 9.9% | 8.3% | 6.4% | 3.3% | 2.1% | 1.6% | 1.1% | 3.2% |

*Table 15.* Ablation results of the number of learnable beats.

| #. Beats | PTBXL-Rhythm | | | PTBXL-Form | | | PTBXL-Sub | | | PTBXL-Super | | | CPSC2018 | | | CSN | | | Average |
|---|---|---|---|---|---|---|---|---|---|---|---|---|---|---|---|---|---|---|---|
| | 1% | 10% | 100% | 1% | 10% | 100% | 1% | 10% | 100% | 1% | 10% | 100% | 1% | 10% | 100% | 1% | 10% | 100% | |
| 10 | *88.83* | 94.65 | 96.91 | 63.41 | 76.71 | 83.30 | 79.22 | 84.40 | **87.46** | **85.82** | *87.61* | *87.87* | 88.54 | 91.75 | **94.32** | 78.25 | 84.83 | 90.17 | 85.78 |
| 12 | 87.98 | 94.73 | 96.81 | 62.33 | 76.94 | 84.35 | 79.69 | 84.99 | 86.86 | 85.61 | 87.57 | 87.79 | *88.58* | 92.70 | 93.76 | **79.89** | 87.22 | 90.29 | 86.00 |
| 14 | 87.12 | 95.82 | 96.53 | *64.11* | **78.92** | **84.80** | 80.30 | **85.98** | *87.31* | 85.39 | 87.40 | 87.66 | 87.58 | *92.84* | *94.14* | *79.11* | **87.87** | **91.50** | *86.35* |
| 16 | **90.37** | **96.68** | **97.32** | **64.74** | 76.91 | 83.21 | **80.66** | *85.17* | 87.03 | 85.56 | 87.48 | 87.49 | **89.18** | **93.15** | 94.07 | 78.91 | 87.18 | 90.23 | **86.41** |
| 18 | 88.01 | *96.48* | *97.20* | 62.71 | *77.51* | *83.98* | *80.37* | 84.99 | 86.88 | *85.68* | **87.63** | **87.93** | 87.65 | 92.75 | 93.74 | 78.48 | *87.23* | *91.25* | 86.14 |

## C.2. Ablation on Number of Beats.

In the main text, we have chosen 10 heart beats is because we assume that the majority coherts in MIMIC-IV-ECG database have 10 heart beats for each ECG recordings. (The sample duration is 10 seconds and normal heart beat is 1 beat per second). However, we have carefully calculate this statistics and found that median number of heart beats are 12 - 13, as evidenced by Table 14. Thus, we have conducted further ablation study to explore more insights of this hyperparameter, and the results are as below in Table 15: The model with 16 learnable heart beats performance best among all variants and have a performance gain of +0.66% compared with 10 learnable beats setting.

## D. Dataset Details.

Table 16 summarizes the key statistics of the datasets used in this study. The MIMIC-IV-ECG dataset provides a large corpus of 760,618 ECG records without explicit diagnostic labels, serving as unlabeled data for pretraining. In contrast, PTB-XL, CPSC2018, and CSN are labeled datasets used for downstream evaluation. PTB-XL comprises 12,978 training, 1,642 validation, and 1,652 test samples, spanning five diagnostic categories such as NORM, CD, and HYP. CPSC2018 includes 8,958 training, 1,303 validation, and 2,598 test samples across nine rhythm- and morphology-related classes, including AF, PAC, and RBBB. CSN consists of 7,651 training, 851 validation, and 2,126 test samples, with annotations for four diagnostic categories such as AF and GSVT. Together, these datasets encompass a broad spectrum of clinically relevant cardiac conditions, enabling robust assessment of model generalization across diverse domains.

## E. Implementation Details

### E.1. Evaluation Metric

*AUC* (Area Under the Curve) quantifies the overall classification performance as the area under the ROC curve. Ranging between 0 and 1, a higher AUC indicates superior model discriminative power. Specifically, it estimates the probability that the model ranks a randomly selected positive instance higher than a negative instance. Our optimization objective prioritizes AUC maximization to achieve optimal TPR-FPR trade-offs.

*Table 16.* Detailed statistics of the datasets.

| Split | PTB-XL | | | | | | CPSC2018 | | | | | | | | | | CSN | | | | |
|---|---|---|---|---|---|---|---|---|---|---|---|---|---|---|---|---|---|---|---|---|---|
| | NORM | CD | HYP | MI | STTC | Total | NORM | AF | I-AVB | LBBB | RBBB | PAC | PVG | STD | STE | Total | AF | GSVT | SB | SR | Total |
| Train | 7254 | 2048 | 1353 | 416 | 1907 | 12978 | 1213 | 1289 | 889 | 243 | 1964 | 864 | 1084 | 1148 | 264 | 8958 | 1583 | 1639 | 2804 | 1625 | 7651 |
| Val | 916 | 234 | 172 | 64 | 256 | 1642 | 197 | 168 | 101 | 33 | 292 | 130 | 146 | 178 | 58 | 1303 | 186 | 189 | 315 | 161 | 851 |
| Test | 913 | 256 | 184 | 56 | 243 | 1652 | 365 | 342 | 251 | 56 | 589 | 274 | 308 | 345 | 68 | 2598 | 449 | 472 | 769 | 436 | 2126 |

*Table 17.* Domain transfer category matching.

| Target Category | PTBXL-Super | CPSC2018 | CSN |
|---|---|---|---|
| AFIB | - | AFIB | AFIB |
| APB | - | PAC | - |
| CLBBB | CD | - | - |
| CRBBB | CD | - | - |
| HYP | HYP | - | RVH, LVH |
| LBBB | CD | - | CLBBB |
| MI | MI | - | MI |
| NORM | NORM | NORM | SR |
| PAC | - | PAC | - |
| SR | - | NORM | SR |
| STD | STTC | STD | STE, STTC, STTU, STDD |
| STE | STTC | STE | STE |
| STTC | STTC | - | STTC, STE, TWO, STTU, QTIE, TWC |
| VPC / VPB | - | VPC | VPB |
| 1AVB | CD | 1AVB | 1AVB |
| 2AVB / AVB | - | - | 2AVB, 2AVB1, AVB |
| RBBB | CD | CRBBB | RBBB |

*Table 18.* Network architecture of MELP.

| Block Name | Layer Number | Layer Components |
|---|---|---|
| ECG Transformer Encoder | 8 | MultiHeadAttention, Dropout, LayerNorm, FC layer, LayerNorm |
| ECG Feature Extractor | 4 | Conv1d, Dropout, Fp32GroupNorm, GELU |
| ECG Positional Encoding | 1 | Conv1d, SamePad, GELU |
| ECG Attentional Pooler | 1 | MultiheadAttention, LayerNorm |
| Text Attentional Pooler | 1 | MultiheadAttention, LayerNorm |
| Text Encoder | 12 | BertAttention, BertIntermediate, BertOutput |
| Text Projector | 1 | Linear, GELU, Linear |
| Text Decoder | 6 | ResidualAttentionBlock, LayerNorm, MultiheadAttention, Identity, LayerNorm, MLP, Identity |

## E.2. Domain Transfer Experimental Details

Table 17 provides the label mappings used for domain transfer experiments. We adopt the SCP-code label alignment protocol as proposed in (Liu et al., 2024a). Specifically, we conduct transfer learning by training on one dataset and evaluating on another, using the aligned target labels to assess generalization under domain shifts. Since MELP supports zero-shot classification, transfer learning is performed by directly evaluating its predictions on the overlapping label set between source and target datasets. Categories without a direct correspondence are excluded from evaluation to ensure consistency and fairness in performance comparison.

## E.3. Network Architecture

Table 18 presents the detailed network architecture of MELP, which comprises modules for both the ECG encoder and the text encoder. The ECG encoder includes an 8-layer Transformer Encoder with multi-head attention, dropout, layer normalization, and feedforward layers, designed to model long-range temporal dependencies. It is preceded by an ECG Feature Extractor composed of 4 blocks, each containing Conv1d, dropout, GroupNorm, and GELU activation. Positional information is encoded using a dedicated Conv1d-based Positional Encoding block. To extract semantically meaningful pooled embeddings, both the ECG and text branches include an Attentional Pooler built with multi-head attention and layer normalization.

The text encoder consists of a 12-layer Transformer, followed by a lightweight Text Projector for dimensionality alignment. Notably, we implement causal attention in the text encoder to prevent information leakage. A 6-layer Text Decoder integrates residual attention blocks and multi-head attention mechanisms to produce cross-modal outputs.

