# OpenReview forum: "From Token to Rhythm: A Multi-Scale Approach for ECG-Language Pretraining"
_ICML.cc/2025/Conference — ICML 2025 poster_

### Official Review · Reviewer_5Gq6 · 2025-03-12

**Overall Recommendation:** 4

**Summary:**

This paper introduces MELP, a novel multi-modal ECG foundation model that leverages hierarchical supervision at the token, beat, and rhythm levels from clinical text to improve ECG representation learning.

Experimental results on multiple public ECG datasets demonstrate that MELP outperforms existing self-supervised and multi-modal models in tasks such as zero-shot classification, linear probing, and transfer learning.

**Claims And Evidence:**

The authors assert that ECG signals have an inherent hierarchical structure with three distinct levels. They argue that clinical text naturally encodes meaningful information corresponding to each of these levels.

This is motivated by the clinical practice where cardiologists first examine fine-grained waveform details (token level), then group these details into individual heartbeats (beat level), and finally assess the overall rhythm (rhythm level).

Although the authors argue that ECG signals possess a hierarchical structure at the token, beat, and rhythm levels, it is difficult to fully embrace token-level learning of ECG and clinical text given that beat-level or lead-level analysis is the standard in ECG interpretation. However, I agree that this approach may reveal novel insights that conventional clinical interpretations might overlook.

**Essential References Not Discussed:**

They referred to recent papers.

**Experimental Designs Or Analyses:**

The authors present a thorough evaluation of their proposed model on multiple publicly available ECG datasets (e.g., PTB-XL, CSN, and CPSC2018) across various downstream tasks.

They conduct comprehensive evaluations and extensive ablation studies comparing different configurations.

The ablation studies involve removing one or more components, with the observed performance drops indirectly confirming the contributions of each level.

However, the paper does not offer a fully isolated evaluation of each hierarchical level (token, beat, rhythm). A more granular, independent analysis of each level could further strengthen the empirical support for the model’s design.

**Methods And Evaluation Criteria:**

The paper introduces a multi-scale cross-modal pretraining framework that aligns ECG signals with clinical text at three hierarchical levels.
At the token level, an encoder-decoder generates report tokens to capture fine-grained waveform details.
At the beat level, token embeddings are aggregated into beat representations and aligned with clinical sentences using contrastive learning.
At the rhythm level, global ECG representations are created by averaging beat embeddings and aligning them with overall text using a global contrastive loss.
Overall, the approach is presented in a way that is largely consistent with the intuitive understanding of ECG analysis.

**Other Comments Or Suggestions:**

-

**Other Strengths And Weaknesses:**

One strength of the paper is its clear and well-written presentation.

However, while the authors acknowledge certain limitations, it would have been interesting to see experiments that vary the number of beat quantizations, as this could potentially yield more intriguing results.

**Questions For Authors:**

As mentioned earlier, it would be beneficial if the paper provided further explanation or experimental results on the following two points:

1. Although token-level learning is an innovative approach, it contrasts with the conventional beat-level or lead-level analysis typically used in ECG interpretation. It would be useful to see additional experiments or detailed explanations on how token-level learning compares with standard methods.

2. The current evaluation assesses the integrated contribution of token, beat, and rhythm levels via ablation studies, where one or more components are removed. However, a more granular, independent evaluation of each level would provide clearer insights into how each contributes to the overall performance. Experiments that isolate token-level, beat-level, and rhythm-level supervision could help verify the specific benefits and potential limitations of each component.

3. Many recent ECG representation learning papers include a reconstruction loss alongside contrastive loss to capture both the generative and discriminative features of the data. The absence of a reconstruction loss in the proposed method may be a deliberate design choice.
While reconstruction loss could help retain fine-grained signal details, it might also add complexity and may not be necessary if the contrastive loss sufficiently captures the critical features.

**Relation To Broader Scientific Literature:**

This paper extends previous multimodal studies that utilize ECG signals and clinical text.

Compared to earlier works, it captures the intuitive perspective of ECG analysis more effectively, and its attempt to evaluate the model fairly using a standardized protocol is particularly noteworthy.

**Theoretical Claims:**

There isn’t a particularly strong theoretical claim.

---

> ### Author Rebuttal · Authors · 2025-04-01
>
> We thank the reviewer for the thoughtful evaluation and for recognizing the strength of our comprehensive experiments and the novelty of our multi-level supervision design. **We also updated the manuscript accordingly.**
>
> **[D.1] Clarification of Token-level Pretraining (Q1)**
>
> Thanks for acknowledging our approach may offer novel insights! From our understanding, token-level embeddings may denote fine-grained ECG features, such as P wave shape, QRS duration, and ST segment changes, by modeling local temporal patterns. Compared with beat-level, it provides more low-level understanding of ECG signals. Although the cardiologists may not explicitly mention token-level features, we think incorporating this level of analysis still encourages deep learning models to better capture the fine-grained characteristics of ECG signals, ultimately improving the model's generalizability.
>
> **[D.2] Ablation on Isolated Variant (Q2)**
>
> Thanks for the valuable comment! As the global rhythm level plays a central role in zero-shot ECG classification, we did not initially report isolated module results. To address this, we conducted additional ablation studies evaluating each level independently. Results are shown in Table D.1. These results show that our full model consistently outperforms each isolated variant consistently, confirming the effectiveness of multi-level supervision.
>
> *Table D.1*
> | Loss |  | PTBXL-Form | | | CPSC2018 | | | CSN | |Average |
> |--------------------------|----------------|------|------|---------------|------|------|--------------|------|------|----------------|
> |                       | 1%   | 10%   | 100%  | 1%   | 10%   | 100%  | 1%   | 10%   | 100%  | |
> | $\mathcal{L}_{\mathrm{LM}}$       | 52.95 | 63.80 | 76.91  | 64.19 | 73.05 | 85.26 | 69.81 | 79.37 | 84.41 | 72.19 |
> | $\mathcal{L}_{\mathrm{Local}}$    |49.81 | 67.82 | 81.41 | 64.18 | 84.08 | 93.17 | 55.89 | 79.77 | *88.79* | 73.88 |
> | $\mathcal{L}_{\mathrm{g}}$        |  57.93 | 72.14 | 82.07 | 78.52 | 87.07 | 92.57 | 75.94 | 82.04 | 86.66 | 79.44 |
> | MELP        | **63.41** | **76.71** | **83.30** | **88.54** | **91.75** | **94.32** | **78.25** | **84.83** | **90.17** | **83.48** |
>
> **[D.3] Analysis on Number of Beat Tokens**
>
> Thanks for your insightful suggestion! Our initial choice of 10 heartbeats was based on the assumption that most ECG recordings in MIMIC-IV-ECG, with a 10-second duration, would contain roughly one beat per second. However, after analyzing the dataset, we found that the median number of heartbeats per recording is approximately 12–13, as shown in Table D.2.
>
> To investigate the impact of this hyperparameter, we conducted an ablation study with varying numbers of heartbeat tokens. The results, presented in Table D.3, show that our model is relatively robust to this variation. Notably, using 14 heartbeat embeddings yields slightly better performance than our initial setting.
>
> *Table D.2*
> | Beat Count | 8     | 9     | 10     | 11     | 12     | 13     | 14     | 15     | 16     | 17     | 18     | 19     | 20     | Others |
> |------------|-------|-------|--------|--------|--------|--------|--------|--------|--------|--------|--------|--------|--------|--------|
> | Frequency  | 18357 | 47635 | 93075  | 112010 | 112424 | 93830  | 74027  | 62150  | 47997  | 24509  | 15987  | 11606  | 8347   | 23493  |
> | Percentage | 2.5%  | 6.4%  | 12.5%  | 15.0%  | 15.1%  | 12.6%  | 9.9%   | 8.3%   | 6.4%   | 3.3%   | 2.1%   | 1.6%   | 1.1%   | 3.2%   |
>
> *Table D.3*
> | #.Beats | | PTBXL-Form | | | CPSC2018 | | | CSN | | Average |
> |--------------------------|----------------|------|------|---------------|------|------|--------------|------|------|----------------|
> |                      | 1%   | 10%   | 100%  | 1%   | 10%   | 100%  | 1%   | 10%   | 100%  | |
> | 10      | 63.41           | 76.71 | 83.30 | 88.54      | 91.75 | **94.32** | 78.25 | 84.83 | 90.17 | 83.48 |
> | 12      |  62.33           | 76.94 | 84.35 | 88.58 | 92.70 | 93.76 | **79.89** | 87.22 | 90.29 |  84.01 |
> | 14      |  64.11      | **78.92** | **84.80** | 87.58      | 92.84 | 94.14 | 79.11 | **87.87** | **91.50** | **84.54** |
> | 16      |  **64.74**       | 76.91 | 83.21  | **89.18** | **93.15** | 94.07 | 78.91 | 87.18 | 90.23 | 84.18 |
>
> **[D.4] Further Discussion (Q3)**
>
> Thanks for your comment! From our perspective, the contrastive learning excels at learning global representations from ECG. However, its ability to capture fine-grained details is relatively limited. To address this, we have incorporated token-level pretraining based on generation to further enhance the model's detailed understanding for ECG representions. Its benefit can be witnessed in Tables 5 and 6 of our original manuscript, by showing better performance in various downstream datasets. While this added complexity may seem significant, we think it is manageable. Our entire model can be pretrained on 4 GTX 3090 GPUs, with a batch size of 64 per device. We have discussed this point in our revised manuscript.

---

### Official Review · Reviewer_VGx7 · 2025-03-12

**Overall Recommendation:** 4

**Summary:**

This paper proposes a multimodal self-supervised pretraining method for paired electrocardiograms (ECGs) and text. This method, MELP, is unique in its use of multi-scale representation learning and supervision by breaking down an ECG signal into hierarchical levels of the full rhythm view, the smaller beat view, and the smallest token view. Comprehensive experiments show gains over state-of-the-art ECG-language models on a variety of interpretation tasks under varying amounts of labeled data, ranging from zero-shot to fully supervised settings.

## Update after rebuttal
I will maintain my original recommendation of acceptance.

**Claims And Evidence:**

Claims appear to be sound, and results are both thorough and convincing at first glance. My only hesitation surrounds the implementation details of baseline methods: were these methods pretrained from scratch on MIMIC-IV-ECG (like MELP, for fair comparison), or were they taken as is (potentially pre-trained on different data) and fine-tuned?

**Essential References Not Discussed:**

There are a few areas where more examples of relevant literature could be mentioned, but these are not critical omissions that change the interpretation of results.

E.g., there are additional examples of contrastive methods for ECG [1], reconstruction-based methods for ECG [2], hybrid approaches for ECG [3], and other relevant vision-language foundation models for ECG [4-6]. Jin et al. [6] is the most relevant, particularly because it conceptualizes beats as “words”, but I am aware that this represents a concurrent work.

[1] Sangha, Veer, et al. "Biometric contrastive learning for data-efficient deep learning from electrocardiographic images." Journal of the American Medical Informatics Association 31.4 (2024): 855-865.

[2] Yu, Han, Huiyuan Yang, and Akane Sano. "ECG-SL: Electrocardiogram (ECG) Segment Learning, a deep learning method for ECG signal." arXiv preprint arXiv:2310.00818 (2023).

[3] Song, Junho, et al. "Foundation Models for ECG: Leveraging Hybrid Self-Supervised Learning for Advanced Cardiac Diagnostics." arXiv preprint arXiv:2407.07110 (2024).

[4] Han, Yu, et al. "Foundation Models in Electrocardiogram: A Review." arXiv preprint arXiv:2410.19877 (2024).

[5] Tian, Yuanyuan, et al. "Foundation model of ECG diagnosis: Diagnostics and explanations of any form and rhythm on ECG." Cell Reports Medicine 5.12 (2024).

[6] Jin, Jiarui, et al. "Reading your heart: Learning ecg words and sentences via pre-training ecg language model." arXiv preprint arXiv:2502.10707 (2025).

**Experimental Designs Or Analyses:**

Experimental design appears sound. Source code has also been provided to aid reproducibility. As mentioned above, my only hesitation concerns whether baseline methods were pretrained from scratch on the same data as MELP or whether they were used “as is”.

**Methods And Evaluation Criteria:**

Yes. This paper leverages large-scale, well-known ECG datasets and evaluation metrics/settings that are consistent with prior literature.

**Other Comments Or Suggestions:**

- Sec 2.1: Title should probably read “ECG Representation Learning” (rather than “Presentation”)?
- Line 107 RHS: “Yu et al.” is repeated – change this to a parenthetical in-text citation
- Equation 7: Comma should go inside equation
- Fig. 1: In the caption, a space is needed before “Token Level”

**Other Strengths And Weaknesses:**

*Strengths*:
- Writing, organization, and presentation are very high-quality and clear to the reader.
- The multi-scale treatment of ECG signals is unique and appears to be beneficial for downstream performance.
- Experiments are thorough, with large-scale pretraining and validation on a variety of datasets and tasks to many relevant competitive baselines. Ablation studies help identify which components are most useful.

*Weaknesses*:
- A few methodological details can be clarified – no obvious weaknesses! Release of source code and model weights will be important to facilitate reproducibility.

**Questions For Authors:**

1.	How specifically were baseline models treated with respect to pretraining? Were they pretrained from scratch on the same data as MELP, or were their weights used “as is” for eventual fine-tuning or zero-shot evaluation? Alternatively, were results in tables ever taken directly from the paper (without the authors running analyses themselves)? Please clarify these details.
2.	Do the authors plan to publicly release the code and model weights?
3.	In Sec 3.2, how specifically is “cardiology-related data” extracted or filtered from PubMed and Wikipedia? Include these details in the Supplement.

**Relation To Broader Scientific Literature:**

This study falls as one of a few recent vision-language models released for multimodal ECG-text representation learning. However, it is unique in its multi-scale treatment of ECG representation learning and superior performance compared to relevant state-of-the-art models.

**Theoretical Claims:**

N/A

---

> ### Author Rebuttal · Authors · 2025-04-01
>
> We sincerely thank the reviewer for the thoughtful evaluation and for recognizing the novelty of our multi-scale ECG-language pretraining approach. We also appreciate your positive remarks on the clarity of our writing and the thoroughness of our ablation studies. Below, we address your concerns regarding the experimental section in detail. **For each point, we have updated our manuscript accordingly.**
>
> **[C.1] Source of Baseline Results (Q1)**
>
> Thank you for your helpful question. The baseline results reported in Tables 2 and 3 are cited from MERL [1], which is a standardized benchmark by pretraining its model along with 10 existing self-supervised approaches on the MIMIC-IV-ECG dataset. To be fair, we strictly followed the same experimental setup to ensure a fair comparison. Specifically, we adopted the same pretraining dataset, dataset splits, preprocessing pipeline, and fine-tuning hyperparameters as provided in the official MERL GitHub repository. To verify the reported results, we reproduced the MERL model using their released code and pretrained weights (as seen in Table C.1). Our reproduced results were consistent with those in the original paper, supporting their reliability. In some cases, our reproduced performance was slightly lower, likely due to differences in hardware or software environments. To ensure fairness, we report the original (higher) results from the MERL paper in our comparisons. We will clarify this point in the revised manuscript and include the updated explanation accordingly.
>
> *Table C.1*
>
> |   |  | PTBXL-Rhythm |  | | PTBXL-Form | | | PTBXL-Sub | | | PTBXL-Super | | | CPSC2018 | | | CSN | |Average |
> |------------------------|---------------------|----------------------|-----------------------|------------------|-------------------|--------------------|-----------------|------------------|-------------------|-------------------|-------------------|--------------------|-----------------|-----------------|------------------|-------------|-------------|------------|------------|
> |   | 1% | 10% | 100% | 1% | 10% | 100% | 1% | 10% | 100% | 1% | 10% | 100% | 1% | 10% | 100% | 1% | 10% | 100%  | Average |
> | Reproduced results | 45.33               | 83.92                | 86.13                 | 56.62            | 66.03             | 76.57              | 71.41           | 79.05            | 83.30             | 81.19             | 84.66             | 86.77              | 61.59           | 80.07           | 88.83            | 62.33       | 80.22       | 83.44      |   75.41   |
> | MERL paper         | 52.33               | 82.88                | 88.34                 | 58.26            | 72.43             | 79.65              | 64.90           | 80.56            | 84.72             | 82.39             | 86.27             | 88.67              | 70.33           | 85.32           | 90.57            | 66.60       | 82.74       | 87.95      |   78.05  |
> | Difference         | 7.00                | -1.04                | 2.21                  | 1.64             | 6.40             | 3.08               | -6.51           | 1.51             | 1.42              | 1.20              | 1.61              | 1.90               | 8.74            | 5.25            | 1.74             | 4.27        | 2.52        | 4.51       |   2.64   |
>
> **[C.2] Release of Source Code and Weights (Q2)**
>
> Thanks for your great suggestion! To support reproducibility, we will release the full codebase and pretrained model weights to facilitate further research in the community.
>
> **[C.3] Curation of Cardiology-related Corpus (Q3)**
>
> Thank you for the comment. We followed the HeartBERT [2] procedure to collect cardiology-related data from PubMed and Wikipedia. For PubMed, we used cardiology journal names and glossaries from SJR, NIH, Aiken, and the Texas Heart Institute to query abstracts via the API. For Wikipedia, we extracted articles under the "Cardiology" category and its subcategories, supplemented with glossary-based queries. This resulted in a curated dataset of ~5.6 GB (912.5M corpus). We have added more details in the Appendix of revised manuscript .
>
> **[C.4] Not Discussed References and Typos**
>
> Thanks for pointing these out! We have modified typos and updated not discussed references in our revised manuscript according to your suggestions!
>
> **[C.5] Clarification about Methodology Details**
>
> Thank you for the positive feedback and helpful suggestions. We have revised the methodology section to improve clarity.
>
> **References**
>
> [1]. Liu et al. Zero-shot ecg classification with multimodal learning and test-time clinical knowledge enhancement. ICML, 2024.
>
> [2]. Gwon et al. Medical language model specialized in extracting cardiac knowledge. Scientific Reports, 2024.

---

### Official Review · Reviewer_DjPw · 2025-03-13

**Overall Recommendation:** 3

**Summary:**

This study proposes MELP (Multi-scale ECG-Language Pretraining), which introduces an innovative multi-scale supervision mechanism in the field of ECG pretraining. By integrating cross-modal alignment at the token, beat, and rhythm levels, MELP effectively enhances the feature learning capability of ECG signals. Compared to existing methods, MELP achieves significant performance improvements in zero-shot ECG classification, linear probing, and transfer learning tasks, demonstrating exceptional generalization ability, especially in low-data scenarios.

**Claims And Evidence:**

The overall argumentation of the paper is relatively clear, and MELP's performance is systematically validated through standardized benchmarks. However, the implementation of token-level pretraining in the paper may not be entirely convincing. The study employs token-level embeddings to predict the masked portion of the corresponding text, yet the text itself provides an overall description of the ECG signal (e.g., "sinus rhythm"). As I understand it, if the goal is to learn fine-grained representations of ECG signals, the corresponding text should also contain fine-grained descriptions. The authors need to further clarify the motivation and experimental design for this aspect.

**Essential References Not Discussed:**

The paper does not cite METS (Frozen Language Model Helps ECG Zero-Shot Learning, published in MIDL 2023), which is a pioneering work in the field of ECG-Text multimodal learning. METS first proposed an ECG-Text multimodal zero-shot learning approach, making it a crucial reference for understanding this study.

**Experimental Designs Or Analyses:**

The experimental setup follows the configuration recommended by MERL and is generally reasonable.

**Methods And Evaluation Criteria:**

Yes

**Other Comments Or Suggestions:**

- In the Related Work section, ST-MEM is incorrectly spelled as ST-EME。
- Incorrect citation of HeartLang. In page 3, Beat view: Heart beat-sentence Alignment, and Table 1, HeartLang is cited inconsistently in two different ways. Please use the latest citation: https://openreview.net/forum?id=6Hz1Ko087B.

**Other Strengths And Weaknesses:**

The originality of this paper lies primarily in its introduction of the multi-scale concept into multimodal pretraining and its subsequent validation of its effectiveness. However, I believe the main weakness of this paper stems from certain motivational issues in token- and beat-level training. The textual reports used in MELP describe the overall ECG signal rather than providing descriptions at the heartbeat or waveform level. This mismatch in granularity may limit the alignment effectiveness of MELP.

**Questions For Authors:**

- The token-level alignment seems to have a granularity mismatch issue. Could you further clarify the motivation behind this design choice?
- How are positive and negative samples selected for the rhythm-level contrastive loss?
- If there is indeed a text granularity misalignment issue, could this limitation be explicitly acknowledged in the paper’s discussion of current constraints?

**Relation To Broader Scientific Literature:**

This paper can be associated with the field of ECG-Text multimodal learning, offering new perspectives for accurate disease diagnosis in the future.

**Theoretical Claims:**

I have checked the theoretical section in the main text and found no issues.

---

> ### Author Rebuttal · Authors · 2025-04-01
>
> We thank the reviewer for the thoughtful review and for acknowledging the novelty of our multi-level supervision and the strength of our experiments. We also appreciate your constructive feedback on the token-level design, which we address below. **All responses are updated in the revised manuscript**.
>
> **[B.1] Clarification of Token-level and Beat-level Supervision (Q1)**
>
> Thanks for your insightful feedback. While some ECG reports summarize rhythm-level findings (e.g., “sinus rhythm”), many contain detailed waveform-level observations. Below are examples from MIMIC-IV-ECG, with fine-grained observations in bold:
>
> -   Sinus rhythm. **Poor R wave progression** – probable normal variant. **Anterolateral T wave changes** may be due to myocardial ischemia. Abnormal ECG.
> -   Sinus tachycardia. **Short PR interval**. Borderline ECG.
> -   Atrial fibrillation. **Extensive ST-T changes are nonspecific**. Abnormal ECG.
> -   Probable accelerated junctional rhythm. **Low QRS voltages in limb leads**. Abnormal ECG.
>
> High-level diagnoses, like "sinus rhythm", still depend on low-level indicators such as P wave consistency and PR interval regularity. Table B.1 illustrates more examples of how clinical interpretations often depend on both global and local ECG features.
>
> Since our token-level pretraining uses a GPT-style objective, rather than masked token prediction, to generate full diagnostic reports from token-level ECG embeddings. By providing full waveform features to the decoder, we allow the model to learn these relationships and generate reports with varying levels of granularity. It may encourage the model to analyze local features and implicitly learn these indicators. Thus, our token-level pretraining design could enhance the model’s ability to learn more generalized, fine-grained ECG representations.
>
> As for beat-level pretraining, it aims to align beat embeddings with corresponding sentences. While general descriptions in sentences may hinder this process, we think that detailed descriptions encourage the model to understand the ECG at a beat-level.
>
> As shown in the ablation studies in Tables 5 and 6 of the original manuscript, both token-level and beat-level pretraining improve the learned ECG representations, which further supports our claims.
>
> We also agree that including more detailed descriptions in the ECG report would further enhance token-level and beat-level pretraining. Please refer to Reply [B.3] for more details.
>
> *Table B.1*
> | Clinical Diagnosis                       | ECG Criteria                                                                                     | Local Feature Mentioned |
> |------------------------------------------|--------------------------------------------------------------------------------------------------|------------------------------------|
> | **Atrial Fibrillation**                  | Irregularly irregular rhythm                                                                     | False            |
> |                                          | No P waves                                                                                       | True                |
> |                                          | QRS complexes usually < 120ms                                                                    | True                 |
> | **Sinus Rhythm**                         | Regular rhythm at a rate of 60–100 bpm                                                           | False              |
> |                                          | Each QRS complex is preceded by a normal P wave                                                  | True                  |
> |                                          | The PR interval remains constant                                                                 | True                |
>
> **[B.2] Details of Positive and Negative Pairs (Q2)**
>
> Thank you for the helpful question. We train with a batch size of 64 per device across 4 GPUs. Each ECG and its paired report form a positive pair, while all other ECG-report combinations within the mini-batch (i.e., 255 pairs) are treated as negatives for contrastive learning.
>
> **[B.3] Further Discussion of Limitation (Q3)**
>
> Thank you for the suggestion. We further clarified motivation of token-level pretraining in Reply [B.1]. Additionally, we agree that incorporating more explicit fine-grained knowledge, such as detailed waveform descriptions for rhythm diagnosis, could provide stronger supervision and better alignment with clinical criteria. For instance, breaking down terms like "sinus rhythm" into their underlying features (e.g., P waves before each QRS, consistent PR intervals) may enhance learning. We have explicitly added this discussion to the revised Limitations section.
>
> **[B.4] Not Discussed References and Typos**
>
> Thanks for the careful feedback. We have corrected typos and not discussed references in the revised manuscript.

---

### Official Review · Reviewer_dLmC · 2025-03-14

**Overall Recommendation:** 2

**Summary:**

The authors propose Multi-scale ECG-Language Pretraining (MELP), which is a two-step process: First is the cardiology language pretraining step, which pretrains a text encoder using cardiology-focused corpus to maximize the language model’s utility for cardiology. Second step is the multimodal pretraining step, which integrates three levels of cross-modal supervision (token, beat, rhythm) for ECG-language pretraining.

**Claims And Evidence:**

In Section 3.3 Motivation, the authors state “Cardiologists interpret ECG signals in a hierarchical manner, analyzing features at multiple scales – from individual waveform components (tokens) to heartbeats (beats) and overall rhythm”. This claim must be supported with appropriate citation from reputable sources, as it is a crucial statement that inspired the proposed model MELP.

**Essential References Not Discussed:**

N/A

**Experimental Designs Or Analyses:**

Yes I checked the validity of experimental designs and analyses, specifically all classification performance results shown in Tables 2-7.  There are no issues.

**Methods And Evaluation Criteria:**

- The proposed method make sense, but the evaluation criteria lacks depth. Specifically, in Section 3.3 Token-view: Learning to Generate Captions, the authors state “Flexibility in adapting to downstream tasks, such as ECG report generation and ECG-based question-answering” as key advantage of using generative pretraining approach. However, no experiments were conducted to utilize this advantage, and therefore it is unclear why this “advantage” is necessary in this current SSL setting that only conducts ECG classification experiments.
- The authors mention in Section 2.1 ECG Presentation Learning “Recent efforts (Oh et al., 2022; McKeen et al., 2024) combine contrastive and generative objectives to develop ECG foundation models”. However, the authors state these methods overlook rich semantic correlations between ECG signals and clinical text reports, because the methods are pre-trained on only ECG datasets. In the Experiments section, these recent efforts are not included in the baseline models used for comparison against MELP. Therefore, the author's statement is speculative and not substantiated with quantitative evaluations.
- In Section 4.1 Implementation Details the authors mention using Wav2Vec 2.0 architecture for the ECG encoder. However, there are no explanations on why the specific architecture is chosen, or comparison between other architectures such as the  CMSC architecture from CLOCS (Kiyasseh et al., 2021) or Wav2Vec 2.0 + CMSC + Random Lead Masking from (Oh et al., 2022, McKeen et al., 2024).
- The authors conduct evaluations on multiple datasets and various evaluation settings (zero-shot, linear probing, cross-domain adaptation), but only on the ECG classification task. This does not truly test the model’s generalizability beyond ECG classification, which is a limitation given the strong generalizability claims of hierarchical representation learning.

**Other Comments Or Suggestions:**

There are minor typos
- Line 148 pretraing -> pretraining
- Line 427 Table 4.3 -> Table 8

**Other Strengths And Weaknesses:**

Strenghts
- The paper proposes a novel multi-scale approach for ECG-Language Pretraining
- The paper is well-written and easy to understand
- The Ablation Study is conducted thoroughly to answer some questions regarding the effectiveness of multi-scale approach and cardiology language pretraining

Weaknesses
- As mentioned in Evaluation Criteria, the experimental section lacks depth
- Some questions remain unanswered such as why only 10 learnable queries are used in the Heart beat-sentence Alignment section

**Questions For Authors:**

- Why were works such as Oh et al., 2022; McKeen et al., 2024 not included in the baseline models for evaluation?
- Why was the Wav2Vec 2.0 architecture chosen for the ECG encoder? Why were architectures such as CMSC (Kiyasseh et al., 2021) or Wav2Vec 2.0 + CMSC + RLM (Oh et al., 2022) not chosen / compared to the Wav2Vec 2.0 encoder?
- Were any additional downstream tasks conducted beyond ECG classification to show the strong generalizability of hierarchical representation learning? (e.g., Patient Identification)
- Why were 10 learnable tokens used in Section 3.3 Beat view: Heart beat-sentence Alignment?

**Relation To Broader Scientific Literature:**

Research on Self-supervised learning (SSL) methods for ECG-related tasks has gained significant attention in recent years due to its potential to enable various downstream tasks without relying on extensive annotated ECG datasets. The contributions of this paper align well with the broader scientific literature.

**Theoretical Claims:**

No. The paper does not present theoretical claims requiring proof verification. The contributions are empirical rather than theoretical, focusing on the effectiveness of MELP compared to other baseline models in ECG classification.

---

> ### Author Rebuttal · Authors · 2025-04-01
>
> Thanks for the thoughtful and detailed review. **All responses have been updated in our revised manuscript**.
>
> **[A.1] Support for Multi-level Motivation**
>
> Thanks for your suggestion. From prior work, we have found several supports for diagnosing ECG by multi-level observations:
>
> -   **Token-level**: Atrial Fibrillation is identified by the absence of P waves and QRS duration <120ms [1].
> -   **Beat-level**: Sinus rhythm requires each QRS complex to be preceded by a normal P wave [2].
> -   **Rhythm-level**: Left Bundle Branch Block diagnosis requires Variable ventricular rate [2].
>
> As such, we think our proposed multi-scale ECG pretraining is reasonable. We have also incorporated more support in our manuscript.
>
> **[A.2] Additional Compared Baselines (Q1)**
>
> Thanks for your comment. As you suggested, we have compared with Wave2Vec 2.0 + CMSC + RLM (Oh et al., 2022) and ECGFM (McKeen et al., 2024) in below Table A.1. As shown, MELP consistently outperforms these two approaches across all experimental settings.
>
> *Table A.1*
> | Method | | PTBXL-Form | | | CPSC2018 | | | CSN | |Average |
> |--------------------------|----------------|------|------|---------------|------|------|--------------|------|------| ------|
> |   | 1%   | 10%   | 100%  | 1%   | 10%   | 100%  | 1%   | 10%   | 100%  | |
> | Wave2Vec 2.0 + CMSC + RLM | 52.72         | 67.81| 80.72| 75.70       | 88.16| 92.61| 65.65  | 78.82| 87.87|    76.67     |
> | ECGFM  | 60.95         | 74.99| 85.54| 82.18       | 89.52| 93.26| 71.51  | 83.17| 88.89|   81.11      |
> | MELP            | **63.41**  |**76.71**|**83.30**| **88.54** |**91.75**|**94.32**| **78.25**|**84.83**|**90.17**| **83.46** |
>
> **[A.3] Justification of ECG Encoder Architecture (Q2)**
>
> Thanks for your suggestion. Our understanding is that CMSC is a patient-specific contrastive training approach, and Random Lead Masking is a data augmentation strategy. Neither serves as a network backbone. We have used Random Lead Masking by default in our implementation (see code). We also tested CMSC by pretraining the ECG encoder before multimodal training (as it cannot be directly applied into our framework). As shown in Table A.2, the CMSC variant consistently underperforms, with an average drop of -3.53%. Therefore, we did not adopt it in the final model.
>
> *Table A.2*
> | ECG Encoder | | PTBXL-Form | | | CPSC2018 | | | CSN | |Average |
> |--------------------------|----------------|------|------|---------------|------|------|--------------|------|------|----------------|
> |                       | 1%   | 10%   | 100%  | 1%   | 10%   | 100%  | 1%   | 10%   | 100%   |
> | Wave2Vec 2.0     | **63.41**     | **76.71** | **83.30** | **88.54**     | **91.75** | **94.32** | **78.25**  | **84.83** | **90.17** | **83.81** |
> | Wave2Vec 2.0 + CMSC      | 62.07         | 75.55 | 82.57  | 80.69       | 88.40 | 92.91 | 71.89  | 81.00 | 87.42 |  80.28  |
>
> **[A.4] Additional Downstream Tasks (Q3)**
>
> To further show the generalizability of MELP, we evaluated it on two tasks: **ECG report generation** and **patient identification**.
>
> We evaluated report generation using the ECGBench dataset (500 samples), comparing it to the 7B PULSE [3] model. As shown in Table A.3, MELP significantly outperforms PULSE, highlighting its strong fine-grained ECG understanding. Moreover, we evaluated MELP on patient idenfitication in PTB-XL (Table A.4), where it achieved the highest Top-K recall. These results further demonstrates MELP's superior generalizability.
>
> We agree that ECG question answering is a promising direction, while this task requires text encoder finetuning and we will include in future work.
>
> *Table A.3*
> | Models   | Size | BLEU-1 | BLEU-4 | METEOR | ROUGE-L | BERTScore F1 |
> |----------|------|--------|--------|--------|----------|---------------|
> | PULSE    | 7B   | 5.12   | 0.83   | 13.76  | 8.15     | 10.96         |
> | MELP | 284M | **13.02**  | **1.87**   | **11.28**  | **18.50**    | **44.08**  |
>
> *Table A.4*
> | Method                       | R@1   | R@5   | R@10  |
> |------------------------------|-------|-------|-------|
> | Wave2Vec 2.0 + CMSC + RLM    | 39.8  | 52.14 | 59.21 |
> | ECGFM                        | 49.18 | 60.70 | 67.76 |
> | MERL                         | 16.12 | 26.32 | 31.74 |
> | MELP                     | **49.67** | **66.12** | **70.89** |
>
> **[A.5] Analysis of Number of Beat Tokens (Q4)**
>
> Thanks for your valuable question. We initially chose 10 heartbeats based on the assumption of one beat per second in 10-second ECGs. Please see Response [D.3] under Reviewer 5Gq6 for more analysis.
>
> **References**
>
> [1]. Mitchell et al. Canadian Cardiovascular Society atrial fibrillation guidelines 2010: Prevention and treatment of atrial fibrillation following cardiac surgery. Can J Cardiol, 2011.
>
> [2]. Mattu et al. Electrocardiography in emergency, acute, and critical care. American College of Emergency Physicians, 2019.
>
> [3]. Liu et al. Teach Multimodal LLMs to Comprehend Electrocardiographic Images. arXiv preprint 2024.

---

### Decision · Program_Chairs · 2025-05-01

**Decision:**

Accept (poster)

**Comment:**

The paper has received mixed reviews but generally on the positive side. The reviewers generally praised the novelty and empirical strength of MELP across multiple datasets. There did remain some key concerns such as limited evaluation beyond classification tasks (e.g., no demonstration of captioning), lack of clarity around the granularity of token-level alignment with text, and some missing baselines like CMSC and ECGFM, which were addressed in the rebuttal. Overall, I believe the paper is a meaningful contribution to the special area.